# Distinct EH domains of the endocytic TPLATE complex confer lipid and protein binding

Klaas Yperman [1,2], Anna C. Papageorgiou[3,10], Romain Merceron[4,5,10], Steven De Munck[4,5], Yehudi Bloch [4,5], Dominique Eeckhout [1,2], Qihang Jiang[1,2], Pieter Tack [6], Rosa Grigoryan [7], Thomas Evangelidis[3], Jelle Van Leene [1,2], Laszlo Vincze[6], Peter Vandenabeele[6,8], Frank Vanhaecke [7], Martin Potocký [9], Geert De Jaeger [1,2], Savvas N. Savvides [4,5✉], Konstantinos Tripsianes [3✉], Roman Pleskot [1,2,9✉] & Daniel Van Damme [1,2✉]

Clathrin-mediated endocytosis (CME) is the gatekeeper of the plasma membrane. In contrast to animals and yeasts, CME in plants depends on the TPLATE complex (TPC), an evolutionary ancient adaptor complex. However, the mechanistic contribution of the individual TPC subunits to plant CME remains elusive. In this study, we used a multidisciplinary approach to elucidate the structural and functional roles of the evolutionary conserved N-terminal Eps15 homology (EH) domains of the TPC subunit AtEH1/Pan1. By integrating high-resolution structural information obtained by X-ray crystallography and NMR spectroscopy with all-atom molecular dynamics simulations, we provide structural insight into the function of both EH domains. Both domains bind phosphatidic acid with a different strength, and only the second domain binds phosphatidylinositol 4,5-bisphosphate. Unbiased peptidome profiling by mass-spectrometry revealed that the first EH domain preferentially interacts with the double N-terminal NPF motif of a previously unidentified TPC interactor, the integral membrane protein Secretory Carrier Membrane Protein 5 (SCAMP5). Furthermore, we show that AtEH/Pan1 proteins control the internalization of SCAMP5 via this double NPF peptide interaction motif. Collectively, our structural and functional studies reveal distinct but complementary roles of the EH domains of AtEH/Pan1 in plant CME and connect the internalization of SCAMP5 to the TPLATE complex.

[1] Department of Plant Biotechnology and Bioinformatics, Ghent University, Ghent, Belgium. [2] VIB Center for Plant Systems Biology, Ghent, Belgium. [3] CEITEC-Central European Institute of Technology, Masaryk University, Brno, Czech Republic. [4] Department of Biochemistry and Microbiology, Ghent University, Ghent, Belgium. [5] VIB Center for Inflammation Research, Ghent, Belgium. [6] Department of Chemistry, X-ray Microspectroscopy and Imaging – XMI Research Unit, Ghent University, Ghent, Belgium. [7] Department of Chemistry, Atomic & Mass Spectrometry – A&MS Research Unit, Ghent University, Ghent, Belgium. [8] Archaeometry Research Group, Department of Archaeology, Ghent University, Ghent, Belgium. [9] Institute of Experimental Botany, Academy of Sciences of the Czech Republic, Prague 6, Czech Republic. [10] These authors contributed equally: Anna C. Papageorgiou, Romain Merceron. ✉email: savvas.savvides@irc.vib-ugent.be; kostas.tripsianes@ceitec.muni.cz; pleskot@ueb.cas.cz; daniel.vandamme@psb.vib-ugent.be

Internalization of membrane proteins is of crucial importance for cell survival as it allows to quickly react to changing environmental conditions. The residence time of integral membrane proteins at the plasma membrane is controlled by internalization signals that are recognized by adaptor protein complexes, which mediate a process named clathrin-mediated endocytosis (CME). The start of CME is marked by an enrichment of cargo proteins and negatively charged phospholipids[1]. In the next stage, adaptor proteins and clathrin are recruited in a timed fashion and accumulate at the site of endocytosis[2]. In the later stages, accessory proteins and additional clathrin molecules are recruited followed by scission of the formed vesicle from the plasma membrane. The main drivers during this process are low affinity protein–protein and protein–lipids interactions, which enable this dynamic process of assembly and disassembly of clathrin-coated vesicles.

In plants, two adaptor complexes play a role during the initiation phase of endocytosis, the Adaptor Protein-2 complex (AP-2) and the TPLATE complex (TPC)[3,4]. AP-2 and TPC likely have independent but also complementary roles in CME[3]. Both protein complexes have a core-complex of four subunit[5], which in the case of TPC is associated with four additional subunits (TWD40-1, TWD40-2, AtEH1/Pan1, and AtEH2/Pan1)[6]. The AtEH/Pan1 proteins are more loosely associated with TPC as they for example do not associate with the other complex subunits when a truncated TML subunit forces TPC into the cytoplasm[3] and they do not co-purify with the complex in *Dictyostelium*[5]. Recent evidence suggests however a similar arrival time of all TPC subunits at the plasma membrane at the onset of endocytosis, preceding clathrin arrival[3,7,8].

The AtEH/Pan1 proteins are the plant homologs of yeast Pan1p, which is known for its role as an activator of ARP2/3-dependent actin dynamics during endocytosis[9–11]. Pan1p was also recently shown to be part of a phosphorylation-dependent mechanism connecting endocytic vesicles, endosomal compartments, and actin dynamics in budding yeast[12]. The Arabidopsis AtEH/Pan1 proteins were recently shown to mediate actin-dependent autophagy in plants[13]. AtEH/Pan1 and Pan1p proteins are both hallmarked by the presence of two EH domains at their N-terminus[3,13].

In animals and yeast, EH domains have been characterized in great detail due to their presence in crucial endocytic proteins like Eps15, REPS1, EHD1, etc.[14–16] SMART[17] and Prosite[18] analysis identified only six EH domains in Arabidopsis compared to eighteen in humans. In Arabidopsis, EH domains are present in the endocytic recycling regulators EHD1 and EHD2, each having one EH domain[19] and in the TPC subunits AtEH1/Pan1 and AtEH2/Pan1, each having two EH domains. Plant EHD1 and EHD2 proteins have been characterized as homologs of human EHD proteins and were suggested to play a regulatory role during plant CME, plant defense, and salt stress[19–21]. In contrast to the essential function of all tested TPC subunits, silencing of EHD1/2 does not result in severely aberrant phenotypes, indicating redundancy or a more specialized function[3,13,19]. In contrast to the single EH domain in EHD proteins, AtEH/Pan1 proteins harbor two EH domains and we asked if both EH domains serve as independent functional modules. We, therefore, set out to characterize the function of the EH domains of the AtEH/Pan1 proteins in Arabidopsis. In this study, we used a multidisciplinary approach to perform a structural and functional side-by-side comparison of both EH domains of AtEH1/Pan1.

## Results and discussion

**Structural characterization of both EH domains of AtEH1/Pan1 reveals a common fold.** AtEH/Pan1 proteins are highly unstructured but three domains can be identified; a coiled-coil domain implicated in dimerization[22] and two N-terminal Eps15 homology (EH) domains (Fig. 1a, Supplementary Fig. 1). Comparing the EH domains within each AtEH/Pan1 protein shows low sequence identity which is in contrast to the high sequence identity when comparing the particular EH domain between AtEH1/Pan1 and AtEH2/Pan1 (Fig. 1a). Therefore, we decided to focus on both EH domains of AtEH1/Pan1 as representatives, which we hereafter name EH1.1 and EH1.2. To structurally characterize both EH domains of AtEH1/Pan1, we expressed recombinant proteins in *E.coli* and purified highly monodisperse samples for X-ray crystallography and NMR (Supplementary Fig. 1). Crystals of EH1.1 diffracted synchrotron X-rays to 1.55 Å resolution and enabled structure determination via molecular replacement (PDB:6YIG) (Supplementary Table 1). EH1.1 consists of two EF-hands connected by two short antiparallel β-sheets (Fig. 1b). Anomalous scattering supports a calcium ion bound in a pentagonal-bipyramidal geometry in the loop of the first EF-hand (Fig. 1d; Supplementary Fig. 1). No anomalous scattering signal was detected in the second EF-hand loop, but electron density consistent with the coordination of a sodium ion was present (Fig. 1c). Attempts to crystallize EH1.2 were unsuccessful, therefore, we obtained structural insights for both EH1.1 (PDB: 6YEU) and EH1.2 (PDB: 6YET), in solution, by NMR Spectroscopy (Supplementary Fig. 1, Supplementary Table 2)[23]. With respect to EH1.1, the NMR and X-ray structures agree very well with a RMSD of 1 Å. When comparing the NMR structures of EH1.1 and EH1.2, the RMSD between both domains is 2.6 Å. In general, the EH1.1 and EH1.2 domains both have a typical EH domain fold with a very conserved hydrophobic core. The major difference between the two domains is at the interaction interface of the N- and C-terminal ends. The hydrophobic core of EH1.1 is shielded by the proline-rich C-terminal loop, while in EH1.2, the proline-rich N-terminal loop takes over this function.

**The second EH domain coordinates two calcium ions.** We were intrigued by the fact that, in contrast to EH1.1, EH1.2 contains two possible calcium-binding sites. This is manifested by a classical calcium-binding motif (DxDxDxxxxxxE) in the second loop of EH1.2 and a second possible coordination motif in the first loop, where the final glutamate, in the canonical calcium-ligation cassette, is substituted by glutamine in several species including Arabidopsis (Supplementary Fig. 1). Experimental assessment by two independent methods, total reflection X-ray fluorescence (TXRF) and inductively coupled plasma-mass spectrometry (ICP-MS), confirmed the presence of two calcium ions in the Arabidopsis EH1.2 domain (Fig. 1h, i). To our knowledge, the ability of EH domains to coordinate two calcium ions has not been described before. To obtain a possible coordination scheme for Arabidopsis, we performed an extensive all-atom molecular dynamics simulation (3 µs, CHARMM force-field) (Fig. 1e–g, Supplementary Fig. 2, Supplementary Data 6). The all-atom molecular dynamics model, within its time and forcefield limitations, suggests that the first aspartate and the presence of an extra water molecule compensate for the incomplete calcium-binding motif in the first loop of EH1.2 and functionally mimics the role of the canonical glutamate in the calcium-binding motif (Fig. 1g, Supplementary Fig. 2). Restoring the first EF-hand loop to a canonical EF-hand (Q382E), in an all-atom molecular dynamics simulation, resulted in a classical arrangement where glutamate coordinates calcium in a bidentate fashion (Supplementary Fig. 2, Supplementary Data 6). We confirmed by NMR that EH1.2 is indeed more sensitive to precipitation upon calcium chelation compared to EH1.1 (Fig. 1j). This is consistent with our findings that EH1.2 coordinates two calcium ions.

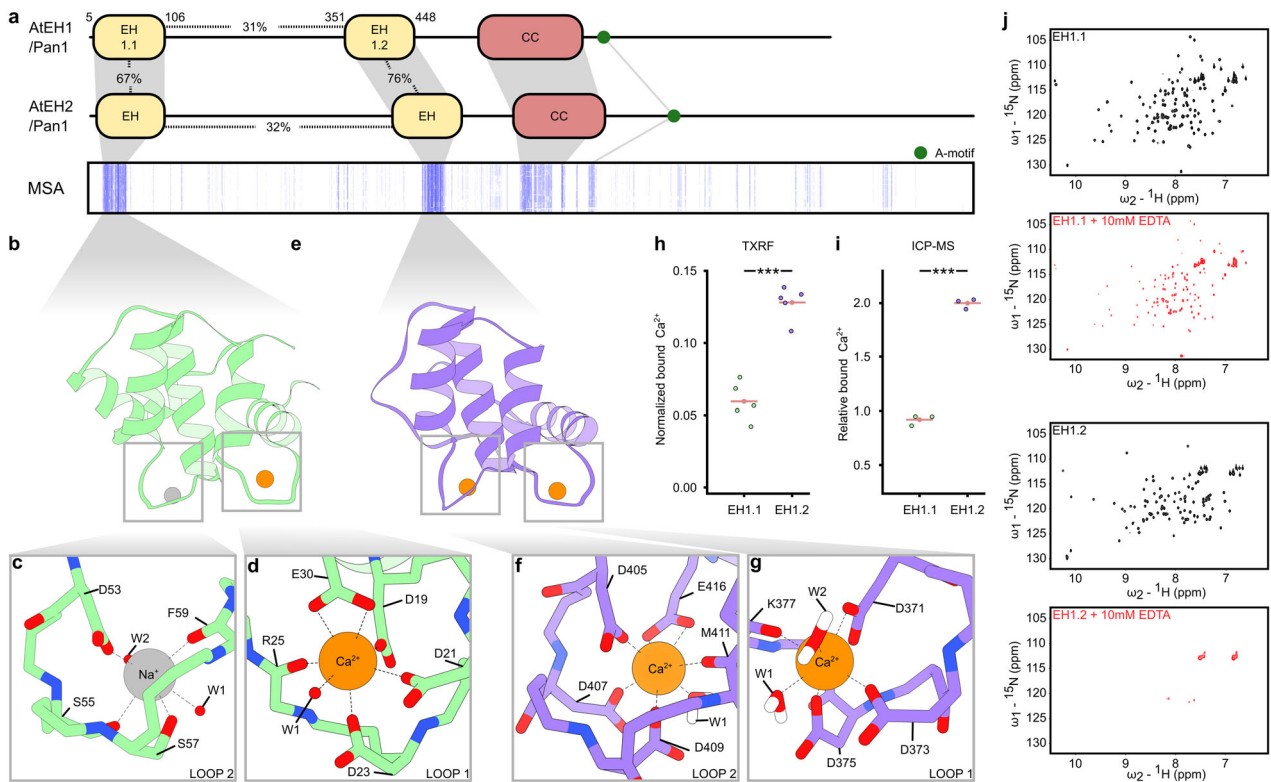

**Fig. 1 EH domains of AtEH1/Pan1 differ in their $Ca^{2+}$-binding capacities. a** Domain organization of AtEH1/Pan1 and AtEH2/Pan1. Both proteins contain two Eps15 homology domains (EH), a coiled-coil domain (CC), and an acidic (A)-motif. A schematic representation of a multiple sequence alignment (MSA), shows strong conservation of the EH domains (blue lines) across the plant kingdom. Percentages indicate the relative number of identical amino acids. **b–g** Cartoon representation of the X-ray structure of EH1.1 and NMR/all-atom molecular dynamics structure of EH1.2. Ions are shown as orange ($Ca^{2+}$) or grey ($Na^+$) spheres. Insets show the ion coordination in each EF-hand loop. $Ca^{2+}$ coordinating residues and water molecules (W) are indicated in (**c**, **d**) and (**f**, **g**). **h** Total reflection X-ray fluorescence (TXRF) intensities of $Ca^{2+}$ normalized to $Cl^-$ of samples containing 2 mM of each EH domain in the presence of 0.5 mM free $Ca^{2+}$. The mean ($n = 5$) is indicated as a pink line. Statistical analysis was performed using a two-tailed Welch's $t$-test ($p = 1.056 \times 10^{-4}$). **i** $Ca^{2+}$ concentration relative to the amount of protein as measured by inductively coupled plasma mass spectrometry (ICP-MS) after subtraction of the amount of free $Ca^{2+}$ in the medium. The mean ($n = 3$) is indicated as a pink line. Statistical analysis was performed using a two-tailed Welch's $t$-test ($p = 1.046 \times 10^{-5}$). **j** $^{15}N$-$^1H$ HSQC spectra of each EH domain before (black) and after (red) $Ca^{2+}$ chelation by 10 mM EDTA.

**AtEH1/Pan1 EH domains differently interact with charged lipids.** EH domains have been proposed to act as a protein interaction hub and/or as a lipid-binding module[24,25]. To unravel the function of the EH domains of AtEH1/Pan1, we tested both domains for their ability to bind peptide motifs and membranes, in a pairwise manner. AtEH/Pan1 proteins mainly function at the negatively charged plasma membrane or the ER-PM contact sites[3,13]. To mimic a negatively charged plasma membrane environment in vitro we performed liposome-binding experiments using an equimolar mixture of phosphatidylethanolamine and phosphatidylcholine (PC) with or without 10% phosphatidylinositol 4,5-bisphosphate (PIP₂). Only EH1.2 substantially bound to PIP₂ enriched liposomes (Fig. 2a, b). Next, we tested the binding of EH1.2 to monophosphate phosphoinositides. We observed only a very weak interaction with phosphatidylinositol 3-phosphate (PI3P) and phosphatidylinositol 4-phosphate (PI4P) compared to PIP₂ enriched liposomes (Supplementary Fig. 3). Given the fact that, on the one hand, calcium is needed for the fold of EH1.2 and, on the other hand, that lipid-EH1.2 interactions are electrostatically-driven, we tested the effect of PIP₂ binding by the EH1.2 domain at different calcium concentrations. We observed increased binding of EH1.2 at lower calcium concentrations (Supplementary Fig. 3). Our results are in line with prior studies showing that high calcium concentrations block the accessibility of charged lipids by shielding and rearranging the lipid headgroups[26].

To understand the difference in the PIP₂-binding of EH1.1 and EH1.2, we compared structural features of both AtEH1/Pan1 EH domains with the EH domains of Human Eps15 (EH2, PDB:1F8H) and EHD1 (EH1, PDB:2KSP), both known for their ability to interact with phosphoinositides[24]. The lipid interacting residues in the EH domains of EHD1 and Eps15 are structurally conserved in EH1.2 (R384, R391, and K398), whereas no lysine or arginine residues are present at those positions in EH1.1 (Supplementary Fig. 3). We generated a triple mutant of the EH1.2 domain (i.e. R384, R391, and K398 were mutated to glutamate, EH1.2 R/K>E) and tested PIP₂ binding (Supplementary Fig. 3). The obtained mutant protein was severely impaired in its interaction with the liposomes containing 10% PIP₂, further supporting that the positively charged residues on the surface are the determinants of the interaction with the PIP₂ molecules (Supplementary Fig. 3).

To show whether the EH domains of AtEH1/Pan1 interact with membranes *in planta*, we prepared them as a triple repeat of each EH domain fused C-terminally with GFP, allowing for an avidity effect. We expressed the three-tandem domains transiently in *N. benthamiana* together with cyclin D. Overexpression of the cyclin D protein triggers cell division in leaves of *N. benthamiana* and thus enables to monitor different stages of cytokinesis in this otherwise differentiated model system[27]. Both three-tandem domains localized to the early cell plate (Fig. 2c), which is the most negatively charged membrane compartment in

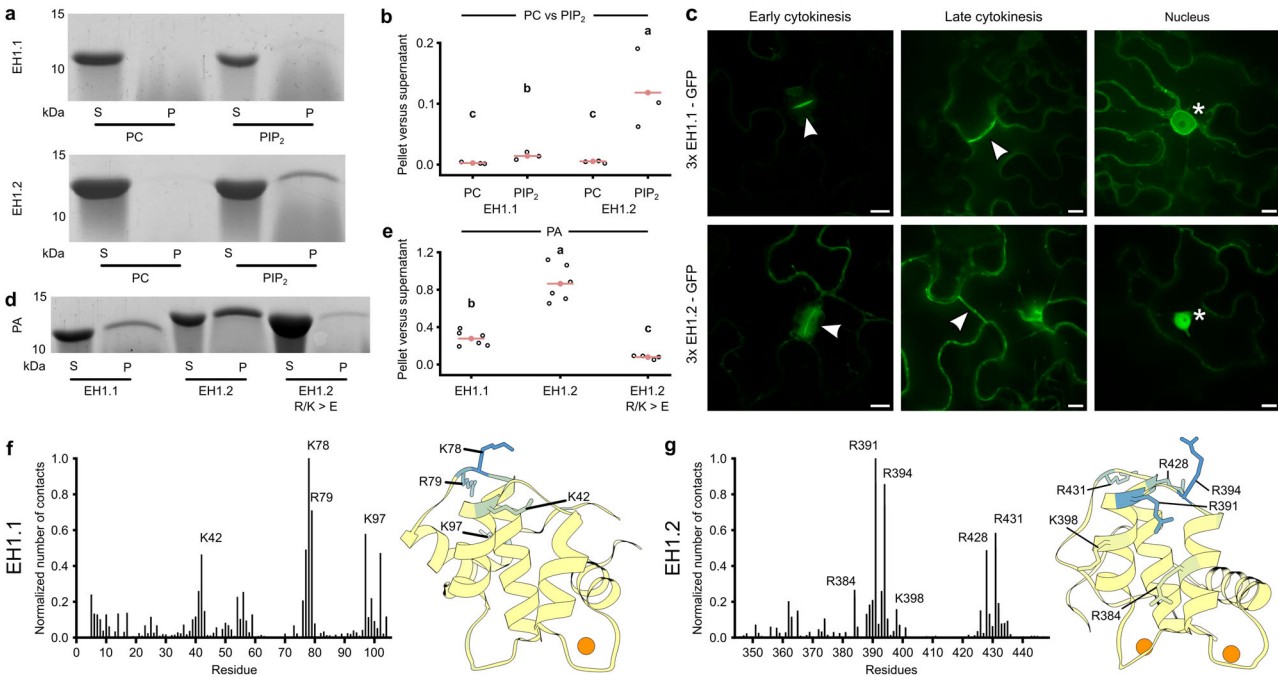

**Fig. 2 Differential binding of EH domains of AtEH1/Pan1 to anionic phospholipids. a** Coomassie-blue stained SDS-PAGE analysis of liposome binding comparing the binding of EH domains between PC (an equimolar mixture of PE and PC) and 10% $PIP_2$ containing liposomes in the presence of 10 μM $Ca^{2+}$. S = supernatant, P = pellet. **b** Quantification of lipid binding (ratio of cumulative grey values between pellet and supernatant) as shown in (**a**). Different letters indicate significant differences between samples using a one-sided Kruskal–Wallis test ($p \leq 0.05$). $n = 3$ independent experiments. **c** Localization of triple EH domains fused C-terminally to GFP in *N. benthamiana* epidermal leaf cells. Cells were triggered to divide by overexpression of the Cyclin D protein. The early and late stages of cell division are depicted (arrowheads). In contrast to EH1.2, EH1.1 is retained more prominent at the late cell plate (arrowhead) and also labels the nuclear envelope (asterisk). Scale bar indicates 10 μm. The experiment was repeated twice independently with similar results. **d** Coomassie-blue stained SDS-PAGE analysis of PA liposome binding of both EH domains as well EH1.2 mutated in its predicted $PIP_2$-binding site (EH1.2 R/K>E). S = supernatant, P = pellet. **e** Quantification of PA lipid binding (pellet versus supernatant) as shown in (**d**). Different letters indicate significant differences between samples using a one-sided Kruskal–Wallis test ($p \leq 0.05$). $n = 4$ or 5 independent experiments. **f, g** CG-MD simulations of EH1.1 (**f**) and EH1.2 (**g**) with a lipid bilayer containing 20% PA. On the left, the mean number of contacts with PA for each domain are shown. The contacts were defined as the number of PA phosphate groups within 0.8 nm of protein atoms calculated through the whole simulation time and averaged over all CG-MD replicas. On the right, a cartoon representation of each EH domain is shown. The color gradient (yellow to blue) indicates the extent of interaction with PA. Residues with the highest contact number are shown as sticks. $Ca^{2+}$ is shown as an orange sphere.

plant cells[28]. In contrast to EH1.2, EH1.1 remained more prominently present at the cell plate in later stages of cytokinesis and also labeled the nuclear envelope (Fig. 2c). The cell plate is devoid of $PIP_2$, but enriched in phosphatidic acid (PA) from very early on[29]. Therefore, we tested a possible direct PA interaction with both EH domains by liposome-binding assays. Consistently with the membrane localization of EH domains *in planta*, we found that both domains directly interact with the liposomes containing 20% PA, with EH1.2 binding PA more efficiently than EH1.1 (Fig. 2d, e).

In order to elucidate the molecular mechanism of the interaction between PA and the EH domains of AtEH1/Pan1, we employed extensive coarse-grained molecular dynamics (CG-MD) simulations. During multiple 1 μs-simulations (in total 35), both domains interacted with the lipid bilayer containing 20% PA (Supplementary Fig. 4). However, we found that the membrane-binding mode of each EH domain substantially differed from each other (Fig. 2f-g). On the one hand, EH1.1 coordinates the PA molecules by three main regions (residues around K42, residues around K78, and residues located at the C-terminus around K97). On the other hand, EH1.2 interacts with the PA molecules predominantly by the region forming the $PIP_2$ binding site (R384, R391, and K398), but additionally, the second region located close to the protein C-terminus (R428 and R431) significantly contributes to the PA coordination. This was corroborated by the liposome-binding experiment with the triple

mutant of EH1.2 in the $PIP_2$ binding site (EH1.2 R/K>E), which showed significantly reduced, but not completely abolished, binding to the PA (Fig. 2d, e). The observed residual PA-binding of the EH1.2 triple mutant is then likely mediated via a second motif, located C-terminally (Fig. 2g). Our data also suggest that, in contrast to EH1.1, EH1.2 coordinates a higher number of PA molecules (Supplementary Fig. 4). The above results are consistent with our biochemical data showing that EH1.2 interacts more efficiently with PA-containing liposomes than EH1.1 (Fig. 2d, e).

A recent report showed that PA significantly contributes to the electrostatic signature of the plant plasma membrane[29] and several plant proteins involved in CME, such as the ANTH domain-containing proteins AP180 and ECA1, were described to interact with PA[30–32]. Our data, showing that the EH domains of the AtEH/Pan1 proteins, which are plant-specific subunits of the TPC[6], also interact with PA thus suggest a more general role of PA in plant CME.

**The first EH domain recognizes a previously unidentified endocytic transport motif in plants.** To identify protein interaction partners of the AtEH1/Pan1 EH domains, we sought to discover interaction motifs. To this end, we digested Arabidopsis seedling proteome and incubated the peptide mix with each EH domain or with GFP as a control (Fig. 3a). Comparative mass spectrometry identified bound peptides. The N-terminal double

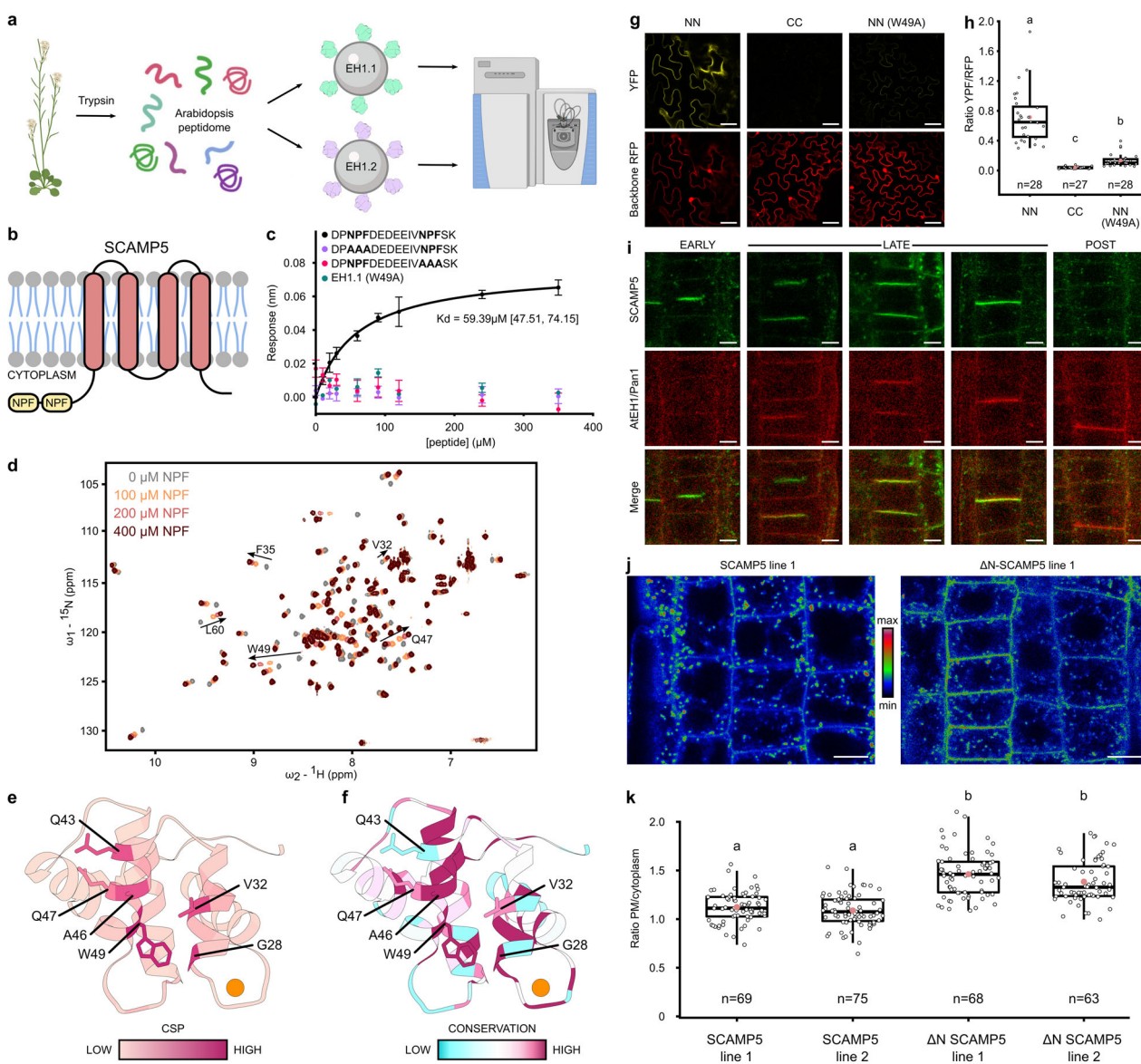

NPF peptide of SCAMP5 was identified as a significant interactor of EH1.1 (Fig. 3b, Supplementary Data 3). This peptide also showed the highest fold change compared to EH1.2 and GFP. Secretory Carrier Membrane Protein 5 (SCAMP5) is part of a five-membered protein family in Arabidopsis. SCAMP proteins were first characterized in mammals for their role in endocytosis[33]. Current insights in the animal and plant field show a broader function of SCAMPs in plasma membrane phase separation, cell plate formation, and pathogen-induced stomatal closure[34–36]. SCAMP5 was also previously identified to reside in close proximity to TPC by proximity labeling[37]. No peptides derived from integral membrane proteins were identified via SMART[17] and Uniprot database[38] searches among the few specific interactors of EH1.2.

Bio-Layer Interferometry (BLI) confirmed the interaction between EH1.1 and a peptide derived from the N-terminus of SCAMP5 with a binding affinity ($K_d$) of 59 μM via steady-state kinetics (Fig. 3c). Both NPF stretches of the N-terminus of SCAMP5 are required to bind EH1.1, as no binding affinity for mutated NPF to AAA mutants could be determined (Fig. 3c). The interaction between the double NPF motif is much stronger for EH1.1 ($K_d^{app}$ ~ 33 μM) compared to EH1.2 ($K_d^{app}$ ~ 190 μM) as

shown by NMR titration experiments (Fig. 3d, Supplementary Fig. 5). NMR peptide titration experiments revealed the binding sites of the peptide on both EH domains (Fig. 3d, Supplementary Fig. 5). Published structures of EH domains interacting with NPF-motifs show the importance of the central tryptophan and its surrounding hydrophobic residues, in agreement with the largest chemical shift perturbations observed in EH1.1[14,15,39] (Fig. 3d, e). Comparison of chemical shift perturbation of EH1.1 with evolutionary conservation across a variety of plant species showed that most of the residues responsible for the NPF motif binding are strongly conserved (Fig. 3f). Mutating the conserved tryptophan to alanine (W49A) in EH1.1, and testing its binding capacity by BLI confirmed its essential role. No binding was observed for the mutated EH1.1 domain (Fig. 3c). We conclude that the N-terminal double NPF motif of SCAMP5 interacts preferentially with EH1.1, mediated via a hydrophobic interaction with the conserved tryptophan residue. The NPF-binding pocket in EH1.1, as identified by NMR chemical shift mapping, has been described before in other EH domains and a similar contribution of the residues surrounding the conserved tryptophan was shown[14,15,39]. In the identified SCAMP5 NPF motif, not one but two NPF motifs are present and separated by an acidic region.

**Fig. 3 The first EH domain of AtEH1/Pan1 interacts with the N-terminal double NPF motif of SCAMP5. a** Scheme of the peptidome profiling experiment. **b** Graphical representation of SCAMP5, the double N-terminal NPF motif is indicated in yellow. **c** BLI steady-state kinetics of the binding of EH1.1 of AtEH1/Pan1 and NPF peptides with or without mutations. WT Protein but not the W49A mutant binds the double NPF peptide with a measurable affinity. Mutation of any of the NPF motifs abrogates binding. Data are presented as mean values ± SD. $n = 3$ independent experiments. **d** $^{1}H$-$^{15}N$-HSQC spectra of EH1.1 titrated with increasing amounts of the SCAMP5 double NPF peptide (grey to red). Peak trajectories of selected residues are indicated by arrows. Their weighted chemical shift perturbations were used to obtain binding isotherms and derive an apparent dissociation constant of the interaction (Supplementary Fig. 5). **e** Cartoon representation of EH1.1 (6-105) colored with a gradient (pink to purple) indicate the extent of chemical shift perturbations induced by the SCAMP5 double NPF peptide binding. Residues showing large chemical shift perturbations (>0.3 ppm) are shown as sticks. **f** Similar as in panel d but the EH1.1 structure is colored according to ConSurf colors denoting evolutionary conservation (blue to white to purple). Most residues affected by peptide binding are well-conserved. **g** Ratiometric bimolecular fluorescence complementation (rBiFC) analysis showing interaction between AtEH1/Pan1 and SCAMP5. Both proteins specifically interact when tagged at the N-terminus. Mutating the SCAMP5-binding site in the first EH domain of AtEH1/Pan1 (W49A) drastically reduces the interaction. Scale bar is 50 μm. **h** Quantification of the YFP/RFP fluorescence ratios from the experiment in (**g**). The black lines represent the median and the red circles represent the mean. The amount of quantified cells is indicated below each boxplot. Letters above the plots indicate statistically significant differences analyzed by one-sided Welch's ANOVA post hoc pairwise comparison was performed with the package multcomp utilizing the Tukey contrasts ($p \leq 0.001$). The boxplot extends from the 25th to 75th percentiles. The line inside the box marks the median. The whiskers go down and up to the 95% percentile. **i** Co-localization of SCAMP5-GFP and AtEH1/Pan1-mRuby3 at the plasma membrane and the cell plate in Arabidopsis root cells. Cells in different phases of cytokinesis (early, late and post) are depicted. SCAMP5 recruitment to the cell plate precedes AtEH1/Pan1 whereas the presence of the latter at the newly formed cross wall exceeds SCAMP5 following completion of cytokinesis. The results shown were observed in two independent lines. Scale bar indicates 5 μm. **j** Confocal analysis of SCAMP5-GFP vs ΔN-SCAMP5-GFP. An increased plasma membrane localization was observed in the absence of the double NPF motif. Colors are shown according to signal intensity (red to green to blue). The images are representative of two independent lines. Scale bar indicates 10 μm. **k** Quantification of the plasma membrane vs the cytoplasm of two independent lines for both constructs as shown in (**j**). The mean is shown as a pink dot. The amount of quantified cells is indicated below each boxplot. Different letters indicate significant differences between samples analyzed by one-sided Welch's ANOVA. Post hoc pairwise comparison was performed with the package multcomp utilizing the Tukey contrasts ($p \leq 0.001$). The boxplot extends from the 25th to 75th percentiles. The line inside the box marks the median. The whiskers go down and up to the 95% percentile.

Double NPF motifs or a single NPF motif flanked by an acidic stretch were identified before in yeast and mammalian systems and these motif extensions were shown to play a role in fine-tuning the interaction with the EH domains[15,16].

Additionally, we confirmed the interaction between AtEH1/Pan1 and SCAMP5 *in planta* using ratiometric bimolecular fluorescence complementation (rBiFC) in *N. benthamiana*. Confocal analysis confirmed the interaction between SCAMP5 and AtEH1/Pan1 specifically when both proteins were fused at their N-terminus with a half of YFP (Fig. 3g, h). The absence of interaction in case of C-terminal fusion proteins fits with the fact that both the EH1.1 domain in AtEH1/Pan1 as well as the NPF motif in SCAMP5 are located N-terminally. To challenge the rBiFC interaction, we repeated the experiment using AtEH1/Pan with a mutated EH1.1 binding pocket (W49A). The interaction was dramatically attenuated with fluorescence ratios that were only slightly higher than the ones observed for the C-terminally tagged proteins (Fig. 3g, h). Comparison of the NMR results on the NPF motif binding by the EH1.1 domain and the membrane-binding mode predicted by CG-MD simulations revealed that the binding sites are not mutually exclusive, i.e., EH1.1 can simultaneously interact with both proteins and lipids (Figs. 2f and 3e). Such a coincidence detection[40] could contribute to fine-tuning of membrane targeting of AtEH1/Pan1 and could further explain the observed differences in the localization of each domain. The nuclear envelope recruitment of the three-tandem EH1.1-GFP construct in *N. benthamiana* (Fig. 2c) could be an example of such a coincidence detection. Recently, PA was shown to reside at the nuclear envelope[41]. EH1.1 may, therefore, simultaneously interact with PA and other double NPF motif-containing proteins, such as the nuclear envelope resident protein NUP98A.

To address the physiological relevance of the interaction between SCAMP5 and AtEH1/Pan1 *in planta*, we imaged *A. thaliana* roots expressing SCAMP5-GFP and AtEH1/Pan1-mRuby3. SCAMP5 localizes mostly in endosomes and weakly at the plasma membrane (Supplementary Fig. 3). Co-localization with AtEH1/Pan1 at the PM was observed. Both proteins also prominently co-localize during various stages of cell plate formation where SCAMP5 clearly precedes the arrival of AtEH1/Pan1. However, the presence of AtEH1/Pan1 at the newly formed cross wall exceeds SCAMP5 following completion of cytokinesis (Fig. 3i). Altogether, our data suggest that SCAMP5 trafficking is highly dynamic. Short-term ES-9 treatment, a potent endocytic inhibitor[42], caused SCAMP5 accumulation at the PM (Supplementary Fig. 6), indicating the endocytic contribution to SCAMP5 dynamics. To further elucidate the role of the AtEH1/Pan1-SCAMP5 interaction in vivo, we compared the localization of the native SCAMP5 protein with an N-terminally truncated version (i.e. lacking the double NPF motif). In comparison to the wild type protein, the ΔN-SCAMP5 showed a reduced endosomal and an increased plasma membrane localization (Fig. 3j, k). These results corroborate our hypothesis that the double NPF motif is a recruitment signal that is involved in the retrograde transport of SCAMP5.

In conclusion, our parallel structural and functional comparison of the EH domains of AtEH/Pan1 revealed two divergent EH domains with differential, yet complementary functions. Both domains interact with negatively charged phospholipids, although they differ in their strength and specificity. In addition, the first EH domain binds a previously unidentified TPC interactor, SCAMP5, via its N-terminal double NPF motif. This constitutes a previously unidentified retrograde transport signal in plants. The tandem of EH domains in AtEH1/Pan1, a recurring leitmotif in the endocytic machinery in Eukarya, is a clear example of the evolutionary division of labor of a repetitive protein fold.

## Methods

**Multiple sequence alignment.** To obtain protein sequences of AtEH/Pan1 homologs, GenBank (https://www.ncbi.nlm.nih.gov/genbank/), Joint Genome Institute (https://genome.jgi.doe.gov/portal/), EnsemblPlants (https://plants.ensembl.org/index.html) and Congenie (http://congenie.org/start) databases were used for a BLASTP search[43]. See Supplementary Data 5 for a complete list of all organisms searched (54 different plant genomes in total). A multiple alignment was constructed with the mafft algorithm in einsi mode[44]. Full sequence alignment can be found in Supplementary Data 5.

**Protein production and purification**. EH domains of AtEH1/Pan1 were amplified from the pDONR plasmid containing the full-length AtEH1/Pan1 coding sequence[3] and cloned by restriction digestion (Nde/Xho) into the pET22b plasmid. Primers used are shown in Supplementary Table 3. The final constructs have an N-terminal His-tag followed by a TEV-protease cleavage site and contain amino acids 1–107 (EH1.1) and 346–449 (EH1.2). To generate the tryptophan mutant in EH1.1, the complete plasmid was amplified using primers over the tryptophan-containing sequence. The fragment was re-assembled using NEBuilder® HiFi DNA Assembly Master Mix (NEB). To generate the EH1.2 R/K>E mutant, the construct was ordered as a gBlocks Gene Fragment (IDT) with (Nde/Xho) cleavage sites (Supplementary Table 4) and cloned into the pET22b plasmid as described above. Constructs were transformed into BL21(DE3) (#C2527H, NEB). Cells were grown at 37 °C in LB⁺ medium and induced by the addition of 0.4 mM IPTG at OD 0.6 for 5 h. The yield of both domains was >5 mg/L culture. To obtain isotope-labeled proteins, cells were grown in M9 minimal medium supplemented with 0.5 g/l $N_{15}H_4Cl$ and/or 2 g/l $^{13}$C-glucose (Eurisotop). Proteins were extracted by sonication in 20 mM HEPES pH 7.4, 150 mM NaCl, 2 mM $CaCl_2$, and Protease inhibitors (cOmplete ULTRA EDTA-free, Roche), except for proteins analyzed by the size exclusion chromatography multi-angle laser light scattering, for which no $CaCl_2$ was added during the purification. Purification was performed on an ÄKTA (GE Healthcare, Unicorn 3.10) system by purification using a HisPrep FF 16/10 (GE Healthcare). A one step elution was performed by adding 500 mM Imidazole during elution. This was followed by a gel filtration step using the same buffer as during the extraction but without the addition of protease inhibitors (HiLoad® 16/600 Superdex® 75 pg (GE Healthcare)). When no His-tag was required the protein was incubated overnight with 1/40 protein:his-TEV-protease (own production) at room temperature without shaking. Uncleaved protein and protease were removed via reverse IMAC (1 ml HisTRAP FF (GE Healthcare) followed by gel filtration (HiLoad® 16/600 Superdex® 75 pg (GE Healthcare)), both in the protein extraction buffer without the addition of protease inhibitors. The protein sequence of EH1.1 and EH1.2 along with their native molecular weight were verified by MS analysis.

GFP-His in an OPINF backbone (a generous gift from the lab of Ray Owen, OPPF, UK) was produced and purified as the EH domains without the addition of $CaCl_2$.

**Multi-angle laser light scattering (MALLS)**. Purified His-tagged proteins EH1.1 (1 mg/ml) or EH1.2 (2 mg/ml) were injected onto a Superdex 75 Increase 10/300 GL size exclusion column (GE Healthcare), equilibrated with 20 mM HEPES pH 7.4, 300 mM NaCl, coupled to an online UV-detector (Shimadzu), a mini DAWN TREOS (Wyatt) multi-angle laser light scattering detector and an Optilab T-rEX refractometer (Wyatt) at room temperature. A refractive index increment (dn/dc) value of 0.185 ml/g was used. Band broadening corrections were applied using parameters derived from BSA injected under identical running conditions. Data analysis was carried out using the ASTRA6.1 software.

**Protein crystallization of EH1.1**. Commercial sparse matrix sitting drop crystallization screens were set up using a Mosquito liquid handling robot (TTP Labtech) using a 100 nl:100 nl, protein (12 mg/ml): mother liquor geometry in SwissSci 96-well triple drop plates. Plates were incubated at 293 K. An original hit in the JCSG screen (1.1 M SodiumMalonate, 0.1 M HEPES pH 7, 0.5% Jeffamine) was optimized to 100 mM HEPES pH 7.6, 0.8 M SodiumMalonate, 0.5% Jeffamine. Crystals were cryoprotected by the addition of ethylene glycol (15% v/v) to the mother liquor prior to plunging the crystals in liquid nitrogen for cryo-cooling prior to data collection.

**Crystallographic structure determination**. X-ray diffraction data were collected from single crystals at 100 K at the P14 microfocus beamline operated by the EMBL at PETRA III synchrotron (Hamburg, Germany). All data were integrated and scaled using XDS (Build20180126)[45]. The initial phases were generated by Automatic Molecular Replacement Pipeline (MoRDa)[46] using a search model derived from the X-ray structure of mouse EHD2 (2QPT) in combination with CCP4 (version 7.0). The initial structure was rebuilt with ARP/wARP[47] and further structure building and refinement was performed using Buster (version 2.10.3)[48] followed by iterative use of COOT[49] (version 0.8.8) and Phenix.refine[50] (version 1.12) software packages.

**NMR structure determination of EH1.1 and EH1.2**. For NMR structure determination the proteins were buffer exchanged using PD-10 columns (Sephadex G-25 M, GE Healthcare) or via gel filtration chromatography in the final purification step to 20 mM MES 6.5, 150 mM NaCl, 2 mM $CaCl_2$. All NMR spectra were recorded at CEITEC Josef Dadok National NMR Centre on 850 MHz Bruker Avance III spectrometer equipped with $^1$H/$^{13}$C/$^{15}$N TCI cryogenic probe head with z-axis gradients. For each protein a set of three sparsely sampled 4D NMR experiments was acquired: 4D HC(CC-TOCSY(CO))NH, 4D $^{13}$C,$^{15}$N edited HMQC-NOESY-HSQC (HCNH), and 4D $^{13}$C,$^{13}$C edited HMQC-NOESY-HSQC (HCCH). Sequential and aliphatic side chain assignments were obtained automatically using the 4D-CHAINS algorithm that combines through-bond information from the 4D-TOCSY experiment and distance information from the 4D-NOESY (HCNH) experiment[23]. Aromatic sidechain frequencies were assigned

manually by recording an additional 3D $^{13}$C edited NOESY-HSQC experiment. Assignment completeness reached 99% for each EH domain. Backbone dihedral angle restraints were derived from TALOS using a combination of five kinds ($H^N$, $H^\alpha$, $C^\alpha$, $C^\beta$, N) of chemical shift assignments for each residue in the sequence[51]. NOE cross-peaks from the three NOESY spectra were assigned automatically by CYANA 3.0 in structure calculations with torsion angle dynamics[52]. Unambiguous distance restraints and torsion angle restraints (Supplementary Table 2) were used in a water refinement calculation[53] applying the RECOORD protocol[54]. The CNS patch introducing calcium coordination in a pentagonal bipyramidal configuration was prepared manually. CNS topology files for calcium coordination were generated based on the high-resolution crystal structure of calmodulin (PDB:1CLL). The quality of the NMR-derived structure ensembles was validated using PSVS[55] (version 1.4).

**All-atom molecular dynamics simulations of $Ca^{2+}$-binding**. All-atom molecular dynamics simulations were performed using the GROMACS 5.1.2 package[56]. We used the NMR structure of EH1.2 containing one calcium atom bound to the canonical motif (loop 2) as a starting structure for the simulation. We added a second calcium atom to the non-canonical motif (loop 1) asking whether it will be stably incorporated in the non-canonical motif and if so, what would be a stable coordination scheme. The same procedure was followed for EH1.2 Q382E. The simulation box contained one EH1.2 or EH1.2 Q382E molecule placed in a cubic box with a length of ~8 nm, which was filled with a 150 mM NaCl aqueous solution and which included additional Cl⁻ ions to neutralize the whole system. The protein and ions were parameterized using the CHARMM36 force field[57]. Water molecules were described with the TIP3P model[58]. Newton's equations of motion were integrated by employing the leap-frog algorithm with a time step of 2 fs. The trajectory frames were recorded every 10 ps. A cutoff of 1.2 nm was applied to short-range electrostatic interactions while long-range electrostatics was calculated with the use of the particle mesh Ewald method[59]. Van der Waals potentials were decreased so that the forces went smoothly to zero between 1.0 and 1.2 nm. Bonds with hydrogen atoms were constrained by the LINCS algorithm[60] and water molecules were kept rigid by the SETTLE algorithm[61]. The temperature of the system was maintained at 310 K using the velocity rescaling thermostat with a stochastic term[62] and the Parrinello–Rahman barostat[63] was utilized for semi-isotropic pressure coupling with a reference pressure of 1.01 bar. The time constants of the thermostat and barostat were 1 ps and 5 ps.

**Coarse-grained molecular dynamics (CG-MD)**. The structure of EH1.1 or EH1.2 was mapped into the MARTINI CG representation using the martinize.py script[64]. The ELNEDYN representation with the distance cutoff 0.9 nm and the spring force constant 500 kJ mol⁻¹ nm⁻² was used to prevent any undesired large conformational changes during CG-MD simulations[65]. The MARTINI CG model for all lipid molecules used in this study was taken from Ingolfsson et al[66]. Lipid bilayer, in total composed of 440 phospholipid molecules, containing dioleoylphosphatidylcholine:dioleoylphosphatidylethanolamine:dioleoyl-phosphatidic acid (molecular ratio 2:2:1) was prepared using CharmmGUI Martini Maker[67]. CG-MD simulations were performed in GROMACS 5.1.2. The bond lengths were constrained to equilibrium lengths using the LINCS algorithm. Lennard-Jones and electrostatics interactions were cut off at 1.1 nm, with the potentials shifted to zero at the cutoff[64]. A relative dielectric constant of 15 was used. The neighbor list was updated every 20 steps using the Verlet neighbor search algorithm. Simulations were run in the NPT ensemble. The system was subject to pressure scaling to 1 bar using Parrinello–Rahman barostat with temperature scaling to 303 K using the velocity-rescaling method with coupling times of 1.0 and 12.0 ps. Simulations were performed using a 20 fs integration time step. Initially, the protein was placed 2.0 nm away from the membrane. Subsequently, the standard MARTINI water, Na⁺ and Cl⁻ ions were added. The final concentration of NaCl was 150 mM. Additional Na⁺ ions were added to the system to ensure the electroneutrality. The whole system was energy minimized using the steepest descent method up to the maximum of 500 steps, and equilibrated for 10 ns. Production runs were performed for up to 2 μs. The standard GROMACS tools, as well as in-house codes, were used for the analysis.

**TXRF**. The EH domains were buffer exchanged using PD-10 columns to a buffer containing 20 mM HEPES pH 7.4, 150 mM NaCl, 0.5 mM $CaCl_2$. The proteins were concentrated to 2 mM and the achieved protein concentration was verified by Nanodrop. TXRF quartz substrate disks were cleaned by placing them in a closed beaker with 5% $HNO_3$ solution under boiling conditions for half an hour. This cleaning process was then repeated using a 3% $HNO_3$ solution, rinsed twice using MilliQ $H_2O$ and a final rinse using a MilliQ $H_2O$—ethanol solution and dried in vacuum. Five replicates for each EH domain consisting of 10 μl of the same protein solution were spotted on a quartz disk and dried under vacuum. The samples were measured in a G.N.R. TX2000 total reflection X-ray fluorescence spectrometer (40 kV, 30 mA, 1000 s LT, Mo anode). XRF data were fitted using the AXIL software package[68]. Ca–$K_\alpha$ integrated intensities were normalized for the Cl–$K_\alpha$ integrated intensities to account for small fluctuations in X-ray tube current and amount of probed sample volume.

**Single-collector sector-field inductively coupled plasma-mass spectrometer (SF-ICP-MS).** For each EH domain, the number of $Ca^{2+}$ ions per molecule was determined via quantification of the calcium concentration in a solution containing a known number of protein molecules, as was determined by Nanodrop (Thermo Fisher). Calcium determination was accomplished using a Thermo Scientific Element XR (Bremen, Germany) SF-ICP-MS.

For this purpose, 10 µl volumes of EH1.1, of EH1.2 and of the buffer solution were diluted with 0.28 M $HNO_3$ (purified by sub-boiling distillation) and measured in triplicate. A buffer containing 150 mM of NaCl, 20 mM of HEPES at a pH of 7.4 and 0.5 mM of $CaCl_2$ was used as a procedure blank for the sample solutions containing 1.98 mM of EH1.1 and 1.94 mM of EH1.2, respectively. Sample preparation was carried out in a class-10 clean lab (PicoTrace™, Göttingen, Germany) at the A&MS-UGent research unit.

The samples were introduced into the ICP ion source using a 200 µl/min quartz nebulizer mounted onto a cyclonic spray chamber. Quantification was accomplished via external calibration. Standard solutions were prepared from 1000 mg/l calcium single-element standard solution (CPI International, Santa Rosa, CA, USA, lot 129901-31). The calibration curve for calcium was generated using five calibration standards, with concentrations ranging from 0 to 50 µg/l. Galium at a final concentration of 10 µg/l (originating from a 1000 mg/l single-element standard solution, Inorganic Ventures, Christiansburg, VA, USA) was used as an internal standard to correct for potential instrument instability and matrix effects. The signals for the nuclides $^{44}Ca$ and $^{69}Ga$ were measured at medium mass resolution (R = 4000) to avoid spectral overlap from $CO_2^+$ at a mass-to-charge ratio of 44. The instrument settings and data acquisition parameters used are summarized in Supplementary Data 1.

**Liposome-binding experiments.** For the liposome binding experiments, a vesicle co-sedimentation assay was used. Vesicle co-sedimentation assay were based on Kooijman et al.[69]. First, lipids were mixed in their specific ratios in chloroform, methanol was also added in the case of PI. Per sample, a total amount of 600 nmol lipids was used. Lipid mixtures were dried in a speedvac and stored at −20 °C. Second, Lipids were redissolved in 500 µl extrusion buffer (250 mM Raffinose pentahydrate, 1 mM DTT, 25 mM Tris-HCl pH 7.5), vortexed and incubated for 1 h after which the samples were sonicated for 1 min. Next, the lipid suspension was extruded twenty times over a polycarbonate membrane (pore size 0.2 µm). Before further use, the suspension was diluted in binding buffer (20 mM HEPES, pH 7.4, 150 mM NaCl and a variable amount of $CaCl_2$ was used) and spinned at $50,000 \times g$ for 30 min at 22 °C. The liposome pellets were resuspended in 150 µl binding buffer. 0.5–1 µg of protein was diluted in 50 µl and added the liposomes. The mix was incubated for 30–45 min at room temperature followed by 30 min centrifugation at $50,000 \times g$. Next, harvest the supernatant. Wash the pellet once by resuspending in 300–500 µl binding buffer, transferring the solution to a new vessel, and spin 20 min at $50,000 \times g$. Resuspend the pellet in 30 µl of the binding buffer and add 10 µl 6× SDS LB. To concentrate the supernatant 1 ml cold acetone (−20 °C) was added and incubated overnight. The next day, the supernatant pelleted by centrifugation at $20,000 \times g$ for 10 min at 4 °C after which the pellets were redissolved in 30 µl of the binding buffer + 10 µl 6 x SDS LB. All samples were incubated at 95 °C for 5 min before loading on gel. Full gels are available in Supplementary Data 4.

**Construction triple EH domains.** Triple EH constructs were designed as a triple-tandem using the same amino acids sequence but variants in codon usage (*A. thaliana*, *Z. Mays*, *N. benthamiana*) separated by a triple GGGS-linker (Supplementary Table 5). At the start and end, 20 bp of pENTRY-L1-L2 plasmid[70] was added to clone the fragment into pENTRY-L1-L2 using the Gibson Assembly master mix (NEB). Next, the obtained plasmid was combined with pEN-L4-pH3.3-R1 and pDONRP2P3-EGFPst into a pFASTRK-II-m43GW backbone[71].

**N. Benthamiana leaf infiltration.** *N. benthamiana* plants were grown in a greenhouse under long-day conditions (6–22 h light, 100 PAR, 21 °C) in soil (Saniflor osmocote pro NPK: 16-11-10 + magnesium and trace elements). Transient expression was performed by leaf infiltration according to Sparkes et al.[72]. A similar optical density of Agrobacterium strains was used for all constructs during co-expression. Transiently transformed *N. benthamiana* were imaged two to three days after infiltration Three-tandem EH domain constructs fused to GFP were expressed together with Cyclin D to induce cell division[27]. Imaging was performed on a PerkinElmer Ultraview spinning-disc system, attached to a Nikon Ti inverted microscope and operated using the Volocity software package (version 6.5.1). Images were acquired on an ImageM ccd CCD camera (Hamamatsu C9100-13) using frame-sequential imaging with a ×60 water-immersion objective (NA = 1.20). Specific excitation and emission was performed using a 488 nm laser combined with a single band pass filter (500–550 nm) for GFP.

**Construction of transgenic plants.** SCAMP5 and ΔN-SCAMP5 cDNAs with LR gateway sites were generated using a BioXP printer (Supplementary Table 4) and cloned using LR clonase (Invitrogen) in pFASTRK-m43GW w/o terminator with a C-terminal GFP. Plant lines were generated by floral dip in Col-0 or complemented AtEH1/Pan1-mRuby3 Arabidopsis lines obtained from previous studies[3]

(Supplementary Table 5). Primary transformants were selected by fluorescent selection of the seeds and the seedlings were checked for expression level on a confocal microscope. T2 lines were used for subsequent experiments.

**Live cell imaging and chemical treatments.** Root epidermal cells of 5–7 day old seedlings, grown vertically on 1/2 MS medium in continuous light were imaged on a Leica SP8X microscope (Leica, software 2019–2020) using the white-light laser with a 40x/1.1NA water-immersion lens.

For sequential dual-color imaging, EGFP was visualized using 488 nm laser excitation and a 495–550 nm spectral detection. mRuby3 was visualized using 558 nm laser excitation and a 600–700 nm spectral detection. Time gating was always applied except for imaging FM4-64.

For ES9 treatment plants were pretreated for 5 min with 10 µM ES9, a generous gift of the lab of Jenny Russinova (PSB, VIB/Ugent, BE), in 1/2 MS media followed by 30 min co-treatment of ES9 with 2 µM FM4-64.

**rBiFC.** rBiFC constructs were created in the 2in1 BiFC vectors[73]. rBiFC clones were amplified from the pDONR plasmid containing the full-length AtEH1/Pan1 coding sequence[3] or the mutated AtEH1/Pan1 coding sequence obtained through Gibson assembly (NEB) or obtained from previous rBIFC experiments[37]. Entry clones were combined in a Gateway LR recombination reaction with an empty BiFC destination vector and selected using LB containing spectinomycin and XgalI. Final BiFC vectors were checked by restriction digestion.

Ratiometric bimolecular fluorescence complementation (rBiFC) images were obtained using a Leica SP8X confocal microscope using the white-light laser with a 40×/1.1NA water-immersion objective. Images were acquired in line sequential mode, using 513 nm excitation and an emission window between 522 and 552 nm for YFP detection and 555 nm excitation and an emission window between 568 and 610 nm for RFP detection. All images were taken using the same ratio between YFP and RFP settings by adapting the strength of the white-light laser. The plasma membrane of the cells was selected by the active contour tool with a diameter of 5 px in ImageJ after which the average intensity of the RFP and YFP channels were selected. All images were devoid of saturated pixels.

**PM/cytoplasm quantification.** For the quantification of plasma membrane versus cytoplasm, the Fiji software package (v1.52p) was used. The 5% most intense pixels of the ROI, with a diameter of 5 px in ImageJ, covering the PM were divided by the 5% most intense pixels of the ROI inside of the cell. Only images devoid of saturated pixels were used for quantification.

**Statistical analysis.** For statistical analysis, the R and multcomp[74] package in R studio (version 1.4.1103) was used.

**Peptidome profiling.** Arabidopsis (Col-0) was grown vertically on 1/2 MS media for 7 days on a nylon mesh, harvested and flash-frozen in liquid nitrogen. Frozen seedlings (3 g/experiment) were ground using mortar and pestle. Proteins were extracted and denatured in 50 mM $NaHCO_3$, 8 M Urea and sonicated three times for 1 min. Protein extracts were rotated for 30 min at room temperature after which the extract was subsequently centrifuged twice at $20,000 \times g$ for 20 min. DTT was added to the supernatant, was added to a final concentration of 5 mM and incubated at 55 °C for 30 min. Proteins were alkylated by the addition of 100 mM iodoacetamide and incubated for 15 min in the dark after which the mix was subsequently diluted with 50 mM $NaHCO_3$ to a final concentration of 2 M urea. Trypsin was added in a ratio of 1:75, protein:trypsin (Sequencing Grade Modified Trypsin, V5117, Promega) and incubated overnight at 37 °C on a rotating wheel.

Trypsin was removed using Sep-Pak Vac 3cc columns (500 mg, WAT036815, Waters). The extract was acidified using 1% TFA for 15 min on ice and cleared by centrifugation at $1780 \times g$ for 15 min at room temperature. The cleared extract was split into four and applied on an equilibrated Sep-Pak columns (Waters, WAT036815, Lot # 010437235B). The column was pre-wet using 5 ml of 100% MeCN followed by sequential washing with 1 ml, 3 ml, and 6 ml of 0.1% TFA. After application of the extract, the column was washed sequentially with 1 ml, 3 ml, and 6 ml of 0.1% TFA followed by a 2 ml wash with 0.1% TFA, 5% acetonitrile. The peptides were eluted using three times 2 ml 0.1% TFA, 40% acetonitrile. The eluate was lyophilized for two days to remove TFA. 200 µg EH domain or his-GFP were coupled, in triplicate, during one hour to 25 µl Ni Sepharose 6 Fast Flow beads (GE Healthcare, 17-5318-01). Unbound protein was removed by three washing steps of 1 ml each. The lyophilized peptides were solubilized in 20 mM HEPES, 150 mM NaCl buffer, 0.2 mM $CaCl_2$, divided and added to the coupled beads. The peptidome was incubated with the proteins for 4 h after which the beads were washed three times with 1 ml of the binding buffer. Peptides were eluted by the addition of 80 µl 20 mM HEPES pH 8, 8 M Urea. The supernatant was removed from the beads and desalted with Monospin C18 columns (Agilent Technologies, A57003100) after which they were freeze dried[75].

Peptides were redissolved in 20 µl loading solvent A (0.1% TFA in water/acetonitrile (98:2, v/v)) of which 5 µl was injected for LC-MS/MS analysis on an Ultimate 3000 RSLC nano LC (Thermo Fisher Scientific, Bremen, Germany) in-line connected to a Q Exactive mass spectrometer (Thermo Fisher Scientific). The peptides were first loaded on a trapping column made in-house (100 µm internal

diameter (I.D.) × 20 mm, 5 μm beads C18 Reprosil-HD, Dr. Maisch, Ammerbuch-Entringen, Germany) and after flushing from the trapping column the peptides were separated on a 50 cm μPAC™ column with C18-endcapped functionality (Pharmafluidics, Belgium) kept at a constant temperature of 35 °C. Peptides were eluted by a linear gradient from 98% solvent A′ (0.1% formic acid in water) to 55% solvent B′ (0.1% formic acid in water/acetonitrile, 20/80 (v/v)) in 30 min at a flow rate of 300 nl/min, followed by a 5 min wash reaching 99% solvent B′. The mass spectrometer was operated in data-dependent, positive ionization mode, automatically switching between MS and MS/MS acquisition for the 5 most abundant peaks in a given MS spectrum. The source voltage was 2.2 kV, and the capillary temperature was 250 °C. One MS1 scan ($m/z$ 400−2000, AGC target $3 \times 10^6$ ions, maximum ion injection time 80 ms), acquired at a resolution of 70,000 (at 200 $m/z$), was followed by up to 5 tandem MS scans (resolution 17,500 at 200 $m/z$) of the most intense ions fulfilling predefined selection criteria (AGC target $5 \times 10^4$ ions, maximum ion injection time 80 ms, isolation window 2 Da, fixed first mass 140 $m/z$, spectrum data type: centroid, intensity threshold $1.3 \times E4$, exclusion of unassigned, 1, 5–8, >8 positively charged precursors, peptide match preferred, exclude isotopes on, dynamic exclusion time 12 s). The HCD collision energy was set to 25% Normalized Collision Energy and the polydimethylcyclosiloxane background ion at 445.120025 Da was used for internal calibration (lock mass).

The raw files were processed with the MaxQuant software (version 1.6.4.0)[76], and searched with the built-in Andromeda search engine against the TAIR10_pep_20101214 database. Parameters can be found in Supplementary Data 2. Intensity values from the peptides output file of MaxQuant were used for quantitative analysis with the Perseus software (version 1.6.1.1). Intensity values were transformed to log$^2$ values. Rows were filtered for at least two valid values in one of the sample groups, GFP, EH1.1, or EH1.2. Missing values were replaced with values from normal distribution with width of 0.3 and a downshift 1.8. To determine the significantly enriched peptide sequences with the EH1 domains, a two-sided Student's $t$-test was performed between each of the EH domains versus GFP and the other EH domain as control. Permutation-based correction for multiple hypothesis testing was performed with thresholds FDR = 0.01 and S0 = 1.

**NMR peptide binding.** NPF peptide was dissolved in the final NMR buffer at a stock concentration of 4 mM. $^1H,^{15}N$ HSQC titrations of $^{15}N$-labeled EH1.1 or EH1.2 with successive addition of unlabeled ligands were performed on samples containing 100 mM protein to a final concentration ratio of 1:4 excess of the peptide. Weighted chemical shift perturbations (CSP) were calculated as:

$$CSP = \sqrt{\delta_{H^N}^2 + \frac{\delta_N^2}{6^2}} \qquad (1)$$

The CSPs were fitted to a binding isotherm using the equation:

$$CSP = \frac{CSP_{max}}{[2P_T]}([L] + [P_T] + K_D - \sqrt{([L] + [P_T] + K_D)^2 - 4[P_T][L]}) \qquad (2)$$

where CSP is the chemical shift perturbation at a given peptide concentration [L], $CSP_{max}$ is the chemical shift perturbation at saturation, $[P_T]$ is the total protein concentration, and $K_D$, the dissociation constant.

**BLI.** BLI experiments were performed on an Octet RED96 instrument (FortéBio, Octet Data acquisition software V12) using an HBS buffer supplemented with calcium (20 mM HEPES pH 7.4, 150 mM NaCl, 2 mM CaCl$_2$). A shake speed of 1000 rpm at 25 °C was used during all measurements. Ni-NTA (Molecular devices, 18-5102) biosensors were functionalized with EH domains (10 μg/ml) till a coupling signal of 3 nm was reached. The proteins were covalently coupled to the Ni-NTA biosensors by sequentially dipping in a 20 mM EDC:10 mM NHS mix for 60 s followed by 60 s quenching in 1 M ethanolamine pH 8.5. The coupled tips were equilibrated in the buffer before the addition of the analyte. Next, functionalized sensors were sequentially dipped in increasing concentrations of analyte with an association time of 60 s and a dissociation time of 120 s. To correct for bulk effects during the measurements we performed double reference subtraction. Here, non-functionalized sensors were exposed to the same analyte concentrations while a functionalized sensor was dipped in zero concentration of analyte. The reference traces were subtracted from the raw data before analysis. Req values used in the analysis were determined for each concentration by averaging 30 data points once the sensors achieved equilibrium. Graphpad Prism 5 was used to analyse and plot binding data using the one-site total binding model. Peptides were ordered from Peptide 2.0 (95% purity).

**Visualisation of protein structures and data.** Bio-Rad Image Lab imager (version 6.0.0 Build 25) was used to scan SDS-PAGE gels. For the visualisation of all protein structures UCSF Chimera (version 1.13.1) was used. Secondary structure of protein models was assigned using the DSSP web server (version 3.0). Mapping of conserved residues was performed by combining the Consurf server[77] with the generated alignment for the individual domains. All figures were prepared utilizing the Inkscape program (https://inkscape.org/).

**Reporting summary.** Further information on research design is available in the Nature Research Reporting Summary linked to this article.

## Data availability

The datasets generated during and/or analyzed during the current study are available as Source data, Supplementary Data files or have been deposited to the Protein Data Bank (pdbe.org). Other data are available from the corresponding author upon reasonable request. The structural data that support the findings of this study are available in the Protein Data Bank with the accession codes 6YIG, 6YEU and 6YET. The following structures were used 2QPT, 1CLL, 2KSP, 1F8H. Source data are provided with this paper.

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

## Acknowledgements

We would like to thank Prof. Eugenia Russinova (Ghent University/VIB-PSB) for an aliquot of the ES9 compound, Prof. Ray Owens (OPPF, Research Complex, Harwell) for initial work on the EH constructs (PID:1724/1846) and an aliquot of the GFP-His plasmid. We also express our gratitude to the proteomics core facility of VIB for the help and expertise with running all MS experiments. This research in the D.V.D. lab is supported by the European Research Council T-Rex project number 682436 and by the National Science Foundation Flanders (FWO; G009415N, 3G017919). The research in the K.T. lab is supported by project CEITEC 2020 (no. LQ1601) with financial contribution from the MEYS CR and National Programme for Sustainability II and by Grant Agency of Masaryk University (MUNI/G/0739/2017). CIISB research infrastructure project LM2018127 funded by MEYS CR is gratefully acknowledged for the financial support of the measurements at CEITEC Josef Dadok National NMR Centre. We thank the staff of beamlines P14 (PETRAIII) and Proxima2A (SOLEIL) for beam time allocation and excellent technical support. Y.B. was supported by a post-doctoral research fellowship from Research Foundation Flanders (FWO, 3E002518, Belgium). R.P. and M.

P. are supported by the Czech Science Foundation grant 19-21758S. S.D.M. was supported by a pre-doctoral fellowship from the Flanders Agency for Innovation and Entrepreneurship (VLAIO-Flanders, Belgium). S.N.S. acknowledges research support from the Hercules Foundation (no. AUGE- 11-029), Ghent University (BOF17-GOA-028) and VIB. Q.J. is supported by the China Scholarship Council (201906760018). Panel a of Fig. 3 was created with BioRender.com. We would like to thank iNEXT (PID: 6554) for funding the structure determination of EH1.1.

## Author contributions

K.Y. and R.P. designed and performed most of the experiments and wrote the paper with S.N.S., K.T., and D.V.D. A.P. performed NMR structure calculations calcium precipitation experiments (NMR) and peptide titration experiments (NMR); T.E. helped with structure calculations; R.M. performed MALLS and X-ray structure determination and helped in designing EH constructs and protein crystallization; S.D.M. performed BLI; Y.B. collected X-ray data and assisted in structure determination; D.E. performed MS analysis; Q.J. helped with characterization of the SCAMP5 lines and rBiFC; M.P. helped in performing liposome binding assays; P.T. performed TXRF; R.G. performed the ICP-MS experiment. K.T., S.N.S., P.V., G.D.J., F.V., L.V., J.V.L., R.P. and D.V.D. were responsible for experimental design, research supervision and finalizing the manuscript text.

## Competing interests

The authors declare no competing interests.
