## [Peer Review File · Nature Communications]

Reviewers' comments:

Reviewer #1 (Remarks to the Author):

In this manuscript, the authors investigate the structure-function of AtEH1/pan1, a member of the TPLATE COMPLEX (TPC), which is an adapter complex involved in clathrin-mediated endocytosis in plants. AtEH contains two Eps15 homology domain (EH domain) of unknown function. By using structural data obtained through a combination of X-ray crystallography, NMR, and molecular dynamics simulation, the authors found that both domains have similar folds except at the interaction interface of the N- and C-terminal ends. The second domain, named EH1.2, coordinates two calcium ions, while other known EH domains coordinate only one. In addition, EH1.2 possesses conserved lysine involved in acidic lipid binding in other EH domain-containing proteins. Accordingly, EH1.2 binds to the negatively-charged phosphoinositide PI(4,5)P2 *in vitro*, but weakly to other phosphoinositides. This interaction is negatively regulated by the addition of calcium in the interaction buffer. *In vivo*, AtEH1/pan1 is localized to the plasma membrane in a manner that is not dependent on PI4P. However, it remains unknown whether PI(4,5)P2 is involved in AtEH1/pan1 localization. The first EH domain (named EH1.1) binds to NPF motifs in the transmembrane protein SCAMP5. SCAMP5 mainly resides in endosomes but chemical inhibition of endocytosis induces its accumulation at the plasma membrane, suggesting that SCAMP5 undergoes endocytosis. A SCAMP5 deletion mutant that lacks its NPF motifs also accumulates at the plasma membrane, which suggests that these motifs are involved in SCAMP5 trafficking. However, this remains unclear whether this is indeed due to interaction with AtEH1/pan1.

The notion that the EH domains in AtEH1/pan1 have divergent functions, perhaps for lipid binding and cargo interaction, is a fascinating concept. This novel idea is supported by convincing *in vitro* evidence and structural data. In addition, the discovery of a novel motif driving endocytosis of plant membrane proteins is an important discovery. However, my enthusiasm is mitigated by the lack of clear *in vivo* evidence for the function of AtEH1/pan1 in SCAMP5 trafficking and the role of lipid binding in AtEH1/pan1 localization/function.

Major points:

- 1) It is unclear whether AtEH1/pan1 (and EH1.2 in particular) binds to PI(4,5)P2 *in vivo* and whether this is important for its localization and function. To assess this the authors could:
 - i) Check the localization of AtEH1/pan1-GFP in mutant backgrounds with altered PI(4,5)P2 levels, such as the pip5k1;pip5k2 double mutant;
 - ii) Analyze the localization/function of AtEH1/pan1 mutated on Lysines 384, 391 and 398 and in parallel test the PI(4,5)P2 binding capacities of such mutants *in vitro* (which would serve as a nice control, as it was done for the tryptophan 49 in the next session)
 - iii) Analyze the localization of the isolated EH1.2 domain to see whether it can localize to the plasma membrane on its own (or in tandem configuration)

- 2) It is unclear whether EtEH1/pan1 is involved in SCAMP5 trafficking and interacts with SCAMP5 *in vivo*. To assess this the authors could:
 - i) Express the W49A AtEH1/pan1 mutant, which does not bind SCAMP5 and test its ability to rescue the pollen phenotype of ateh1 mutant.
 - ii) Verify the interaction between EtEH1/pan1 and SCAMP5 but not SCAMP5deltaN *in vivo*,

for example using co-immunoprecipitation.

Minor points:

- Is the NPF motif conserved in other SCAMP proteins? is it found in other proteins?

Reviewer #2 (Remarks to the Author):

In this work, Yperman, et al. describe structural and biochemical analysis of two EH domains within the AtEH1/Pan1 subunit of the plant-specific TPLATE complex (TPC). The authors of the manuscript are the foremost experts in the TPLATE complex, having isolated the complex and characterized its function in several previous publications. In this work, they investigate the role of two EH domains which seem to have diverged in their specific function without losing their EH structural fold, an interesting insight into the evolutionary diversification of ancestral protein motifs. While TPC has two highly homologous AtEH subunits (AtEH1 and AtEH2), the authors focus on the two EH domains in AtEH1, which they term EH1.1 and EH1.2. Previous NMR structures of EH1.1 and EH1.2 exist and the authors build on this work by determining a high-resolution crystal structure of EH1.1 and using integrative modeling of EH1.2. They show that EH1.1 chelates a sodium ion in addition to the canonical calcium ion, whereas the EH1.2 chelates two calcium ions, a novel finding as all known EH domains chelate a single metal ion. The rest of the work focuses on understanding the diversification of function between the two domains. Using lipid binding assays, the authors propose that EH1.2 has a preference for binding PI(4,5)P2, a lipid enriched at the plasma membrane, consistent with the role of TPC during endocytosis. Using mass spec, binding assays, and fluorescence microscopy, they show that EH1.1 interacts with the SCAMP5 protein and demonstrate that perturbing this interaction shows disruptions in localization and trafficking consistent with the role of TPC in recycling SCAMP5 from the plasma membrane. I think the model is quite intriguing that a single protein fold has been diversified in TPC to lend localization/lipid binding specificity to one domain and cargo recruitment to another.

Overall, the manuscript is well written and the figures and data presentation are superb. Much of the work is excellent and very neatly supports their proposed models. They present many redundant forms of evidence for the metal binding capacity of the two domains and several experiments investigating the SCAMP5 interaction. This work is all technically sound and I have very few comments below in the "minor criticisms" sections. My main criticism is in regard to the lipid binding profile of the two domains. This is a central finding of the work and needs to be addressed before publication.

Major criticisms that need addressing:

The major potential issue in this work is that one of the primary results, namely that EH1.2 has a preference for PI(4,5)P2 binding, is not well supported by their data and in my opinion the two assays they used to come to this finding show contradictory results. The authors claim that EH1.2 and not EH1.1 has a preference for PI(4,5)P2 and that this is perturbed by calcium concentration, suggesting a charged ionic interaction. Consistent with this, EH1.2 has a "basic patch" that is lacking in EH1.1.

In Supplemental Figure 4b, the authors show binding assays using purified EH domains and PolyPiPosomes containing various lipid species (PC, PIP3, PIP4, PI4,5P2) that were

purchased from a company. By eye, the binding preference for both EH domains seems very weak, if non-existent. Both domains appear to bind PC and all PIPs equally well, suggesting there is no discrimination by a single EH domain for a preferred lipid, or a difference in binding profile between the two EH domains. However, simply looking at the gels is not sufficient. The authors performed this experiment in triplicate and quantified the results, normalizing to the PC binding, shown in Supp Figure 4C. No statistical analysis seems to have been performed to show a significant difference between any of the various interactions, and looking at the raw values they have plotted I am not confident that any would be found. With this in mind, I don't think the PolyPiPosome data can be used to make any claims about lipid preference, and maybe even supports the interpretation that there is no lipid discrimination between the EH domains or for a single EH domain for a specific lipid. One major caveat of the above experiments was that they were performed in the presence of 100 uM calcium, which the authors show in another set of experiments weakens the affinity of EH1.2 for charged lipids.

The results that the authors show in the main text in Figure 2 are much clearer by eye. Figure 2A shows that in 10 uM calcium, EH1.2 binds to PI(4,5)P2 and not PC, and EH1.1 does not bind either. Figure 2B shows that in the presence of 100 uM calcium, EH1.2 only binds PI(4,5)P2 and not PC or other charged PIPs (PIP3 and PIP4). Figure 2C shows that EH1.2 binding to PI(4,5)P2 is dependent on calcium concentration. Taken on their own, the results in the main figures seem quite clear and support the claim that EH1.2 and not EH1.1 has a preference for the plasma-membrane enriched PI(4,5)P2. However, these experiments do not seem to be done in triplicate or replicated in the presented work. Considering that the PolyPiPosome assays seem to have variance gel-to-gel, I don't think that a single gel can be taken to support the conclusion of the lipid binding findings.

As the authors note, perturbing PI(4,5)P2 levels in planta is not feasible. They use multiple methods to show that PIP4 does not drive AtEH1 localization (Fig 2F-H), but I believe this data has to stand on its own and can't be used to justify any claims about PI(4,5)P2 binding.

On page 4, the authors claim "PolyPiPosome assays confirmed our findings (Supplementary Figure 4)...". I do not agree with this assertion and, on the contrary, I think that data contradicts the findings from the main figures.

Minor criticisms:

- In Supplementary figure 1, panel e, there is a region of positive difference density that is close to the bound calcium ion. I could not find a reference to this in the text. Is this an unmodeled water? A bound molecule from the mother liquor?

- In Figure 3f the authors describe the colocalization of their proteins of interest during different phases of the cell cytokinesis.

"Both proteins also prominently co-localize during various stages of cell plate formation where SCAMP5 clearly precedes the arrival of AtEH1/Pan1. However, the presence of AtEH1/Pan1 at the newly formed cross wall exceeds SCAMP5 following completion of cytokinesis (Figure 3, panel f)"

As non-experts in plant biology, it took us a good deal of concentration to understand the figure. I think labels for the cells as "early" "mid" or "late/post-cytokinesis" would be helpful. At first glance it looks like multiple examples of the same phenomenon but from the legend it seems that the panels left to right are meant to imply a temporal sequence of cell plate growth throughout cytokinesis.

Reviewer #3 (Remarks to the Author):

The focus of this manuscript is the characterization of two EH domains that belong to the AtEH/Pan1 protein from Arabidopsis. AtEH/Pan1 is a subunit of the TPLATE complex that is involved in clathrin-mediated endocytosis in Arabidopsis. The authors' major conclusion [and statement of significance/novelty] is that the two EH domains, EH1.1 and EH1.2 have different functional properties: one is a sensor of the anionic phospholipid PtdIns(4,5)P₂, and the other interacts with the NPF motif of the novel TPC interacting partner, SCAMP5. In this reviewer's opinion, there are three major problems with this manuscript:

1. Conclusions are not supported by the data
 2. The figures do not support the statements made in the text
 3. The novelty of the work is unclear, as the findings are not at all put in the context of what is known about the EH domains from other organisms (Mouse EHD2 (2QPT) was used for molecular replacement). This pertains to both structural and functional work.
- Issues 1 and 2 are expanded upon below:

A. Results and Discussion section:

"The second EH domain coordinates two calcium ions"

The authors imply that Ca²⁺ is needed for folding. NMR spectra that they show in Fig 1 (i-j) suggest exactly the opposite – addition of 10 mM EDTA to sequester Ca²⁺ has very little effect on the protein conformation. It is therefore unclear how NMR data are used to support the folding argument.

Neither MD simulations nor sensitivity of EH1.2 to precipitation upon sequestration of Ca²⁺ by EDTA can be used to validate the Ca²⁺ binding stoichiometry arguments. The stoichiometry argument (EH1.2 binding two Ca²⁺ ions, not one) – especially if the authors present this as a novel finding – must be validated using direct methods, such as, e.g., inductively coupled plasma measurements.

The NMR structure of EH1.2 appears to be calculated with just one Ca²⁺ ion, and then the second Ca²⁺ ion is added based on the MD results; this structure is presented in Fig 1e. This reviewer questions the validity of this procedure and representation of the structural data, especially given that the Ca²⁺ stoichiometry must be established experimentally using direct methods (see above).

Finally the authors make an argument that Ca²⁺ is required for folding and issue the following statement: "The ability to unfold and refold, relating to a non-functional versus a functional state, in a calcium-dependent manner, hints at a modulatory role for calcium to control the function of this domain." The precipitation that they observe upon removing Ca²⁺ can be a consequence of protein aggregation without loss of fold. The gel-filtration data authors present can also indicate aggregation without unfolding. If the authors wish to make an argument about the loss of fold, they should provide direct evidence that the protein loses the tertiary and secondary structure upon Ca²⁺ removal.

B. Results and Discussion section

"The second EH domain interacts with charged lipids"

All lanes in liposome-binding experiments appear essentially identical. This applies to three sets of data shown in Fig. 2a,b, and e. This reviewer cannot find any support for the

statements issued by the authors:

“Only EH1.2 bound to PI(4,5)P2 enriched liposomes (Figure 2, panel a).”

“We observed a very weak interaction with PI3P and PI4P compared to PI(4,5)P2 liposomes (Figure 2, panel b).”

“We observed increased binding of EH1.2 at lower calcium concentrations (Figure 2, panel e).”

C. Results and Discussion section

"The first EH domain recognises a novel retrograde transport motif"

The authors state “This peptide also showed the highest fold change compared to EH1.2 and GFP (Figure 3, panel a)”. Figure 3a shows a cartoon representation of SCAMP5 but no data.

The authors demonstrate that both domains EH1.1 and EH1.2 (Figure S6) bind the NPF peptides with about 6-fold difference in affinities. Yet they somehow conclude that only EH1.1 recognizes the retrograde transport motif. This conclusion is not supported by the data.

Reviewer #4 (Remarks to the Author):

Comments to the Author

This manuscript reported the structural and functional roles of the evolutionary conserved N-terminal Eps15 homology (EH) domains of the TPC subunit AtEH1/Pan1 in Arabidopsis, which is involved in Clathrin-mediated endocytosis (CME) process. The authors integrated various experimental techniques including X-ray crystallography and NMR spectroscopy with all-atom molecular dynamics simulations, to discover that the first EH domain binds SCAMP5 (a novel TPC interactor) by interacting with the N-terminal double NPF motif of the latter; the second EH domain interacts with negatively charged phospholipids PI(4,5)P2. The complementary roles of the EH domains of AtEH/Pan1 in plant addressed in this manuscript would provide insights into the understanding of CME process in plant. I would like to recommend this manuscript to be published in Nature Communications after a revision.

Specific comments and questions:

1. The Authors claimed that “The all-atom molecular dynamics model, within its time and force field limitations, suggests that the first aspartate and the presence of an extra water molecule compensate for the incomplete calcium-binding motif in the first loop of EH1.2 and functionally mimics the role of the canonical glutamate in the calcium-binding motif” , but they did not provide any evidence, such as structure snapshot in the simulated trajectories or other statistical information from the trajectories for EH1.2 WT to support it. Meanwhile, it seems the simulations were performed in explicit solvent, then, what the water model used in the simulations should be addressed.
2. Supporting Figure 5 was not mentioned in the main text. Further, the specific panel(s) of Supporting Figure should be clarified to make the manuscript more readable.
3. There are some errors or conflicts in the manuscript:
(1) In the first paragraph of page 3, should “With respect to EH1.2, the NMR and X-ray

structures agree very well with an RMSD of 1Å” be “With respect to EH1.1,” ?

(2) In the first paragraph of page 4, should “The lipid interacting residues in the EH domains of EHD1 and Eps15 are structurally conserved in EH1.2 (K391 and K398), we hypothesize a third residue, K384,” be “EH1.2 (R391 and K398)R384”? Or else, it is not consistent with Figure 2c and 2d. Meanwhile, in Figure 2d, for EH1.2, the label “R398” should be “K398”.

(3) The color scale in Figure 3g is not consistent with that in Supporting Figure 7a.

Reviewers' comments: Reviewer #1 (Remarks to the Author):

In this manuscript, the authors investigate the structure-function of AtEH1/pan1, a member of the TPLATE COMPLEX (TPC), which is an adapter complex involved in clathrin-mediated endocytosis in plants. AtEH contains two Eps15 homology domain (EH domain) of unknown function. By using structural data obtained through a combination of X-ray crystallography, NMR, and molecular dynamics simulation, the authors found that both domains have similar folds except at the interaction interface of the N- and C-terminal ends. The second domain, named EH1.2, coordinates two calcium ions, while other known EH domains coordinate only one. In addition, EH1.2 possesses conserved lysine involved in acidic lipid binding in other EH domain-containing proteins. Accordingly, EH1.2 binds to the negatively-charged phosphoinositide PI(4,5)P₂ *in vitro*, but weakly to other phosphoinositides. This interaction is negatively regulated by the addition of calcium in the interaction buffer. *In vivo*, AtEH1/pan1 is localized to the plasma membrane in a manner that is not dependent on PI4P. However, it remains unknown whether PI(4,5)P₂ is involved in AtEH1/pan1 localization. The first EH domain (named EH1.1) binds to NPF motifs in the transmembrane protein SCAMP5. SCAMP5 mainly resides in endosomes but chemical inhibition of endocytosis induces its accumulation at the plasma membrane, suggesting that SCAMP5 undergoes endocytosis. A SCAMP5 deletion mutant that lacks its NPF motifs also accumulates at the plasma membrane, which suggests that these motifs are involved in SCAMP5 trafficking. However, this remains unclear whether this is indeed due to interaction with AtEH1/pan1.

The notion that the EH domains in AtEH1/pan1 have divergent functions, perhaps for lipid binding and cargo interaction, is a fascinating concept. This novel idea is supported by convincing *in vitro* evidence and structural data. In addition, the discovery of a novel motif driving endocytosis of plant membrane proteins is an important discovery. However, my enthusiasm is mitigated by the lack of clear *in vivo* evidence for the function of AtEH1/pan1 in SCAMP5 trafficking and the role of lipid binding in AtEH1/pan1 localization/function.

Major points:

1) It is unclear whether AtEH1/pan1 (and EH1.2 in particular) binds to PI(4,5)P₂ *in vivo* and whether this is important for its localization and function. To assess this the authors could: i) Check the localization of AtEH1/pan1-GFP in mutant backgrounds with altered PI(4,5)P₂ levels, such as the pip5k1;pip5k2 double mutant; ii) Analyze the localization/function of AtEH1/pan1 mutated on Lysines 384, 391 and 398 and in parallel test the PI(4,5)P₂ binding capacities of such mutants *in vitro* (which would serve as a nice control, as it was done for the tryptophan 49 in the next session). iii) Analyze the localization of the isolated EH1.2 domain to see whether it can localize to the plasma membrane on its own (or in tandem configuration).

We agree with the reviewer that the role of the EH domains for AtEH/Pan1 membrane recruitment *in vivo* represents a valid question. However, we believe that in mutant backgrounds with altered PI(4,5)P₂ levels as the reviewer suggests, AtEH/Pan1 proteins might still be recruited to the PM via other lipids (as our new data shows, see below). We also believe that addressing the functionality of the lipid binding capacity of AtEH1/Pan1 *in planta*, using a construct mutated in its lipid binding residues, is not feasible. The respective single mutants in AtEH1/Pan1 and AtEH2/Pan1 are both male sterile. Mutating the proposed residues therefore will in our opinion not allow drawing any conclusions. If the resulting protein fails to complement, this will not allow addressing whether this is caused by failure to bind phosphoinositides or whether the protein is dysfunctional.

On the other hand, if the mutated AtEH/Pan1 proteins remain partially functional, these proteins are part of a multimeric complex, which solely functions at the PM. In a recent study from our

group (Yperman et al., Science Advances, *in press*) we show that, next to the EH domains in both AtEH1/Pan1 and AtEH2/Pan1, at least two other domains, the TML μ HD and the anchor domain of TPLATE can bind negatively charged lipids. The fact that these proteins function in complex with other membrane-binding proteins makes it therefore very difficult to address their lipid binding capacities *in planta* using full-length proteins.

Therefore, as an alternative, we expressed EH1.1 and EH1.2 as a triple repeat in tandem orientation fused C-terminally with GFP in *N. benthamiana* leaves. Co-expression of the three-tandem domains together with cyclin D, which allows to monitor cell plates in this system, resulted in a clear localization to the membrane of the cell plate in case of both EH1.1 and EH1.2 (Figure 2). The early cell plate recruitment of both domains *in planta* inclined us to further investigate the cause of binding. We therefore tested other phospholipids that are present at early stages of the cell division, particularly phosphatidic acid (PA). We found that both EH domains are capable of PA binding *in vitro*, although, similar to our observations with PI(4,5)P₂, they again differed in the strength of their interaction.

In addition, as suggested by the reviewer, we also tested the lipid binding capability of the EH1.2 domain upon mutating the predicted lipid binding amino acid residues. In the presence of 10% PI(4,5)P₂ or 20% PA, a strong abrogation of binding was observed in our liposome binding assay (Figure 2 and Supplementary figure 3).

To gain a molecular understanding of the interaction of both EH domains with PA, we performed coarse-grained molecular dynamics simulations. Consistently with our new biochemical data, the extensive computational modelling predicts that both domains can interact with a lipid bilayer containing 20% PA, although they substantially differ in their membrane-binding mode (Figure 2 and Supplementary figure 4).

2) It is unclear whether AtEH1/Pan1 is involved in SCAMP5 trafficking and interacts with SCAMP5 *in vivo*. To assess this the authors could: i) Express the W49A AtEH1/pan1 mutant, which does not bind SCAMP5 and test its ability to rescue the pollen phenotype of AtEH1/Pan1 mutant. ii) Verify the interaction between AtEH1/pan1 and SCAMP5 but not SCAMP5 Δ N *in vivo*, for example using co-immunoprecipitation.

We thank the reviewer for this comment and we agree that an *in vivo* confirmation of the AtEH/Pan1-SCAMP5 interaction would indeed strengthen our study. Therefore, we opted for ratiometric bimolecular fluorescence complementation (rBIFC) to address the interaction between SCAMP5 and AtEH1/Pan1 *in planta*. We verified the interaction in *N. benthamiana* by comparing N- and C-terminally tagged fusions. Interaction only occurred when both proteins were N-terminally tagged. This fits nicely with our other data, which shows that the interaction occurs between the EH1.1 domain and the double NPF motif, both of which are located N-terminally. Mutation of W49 to A, which is the residue causing the biggest shift upon peptide binding in NMR (Figure 3) in AtEH1/Pan1, lead to a drastic reduction of the interaction, and quantification resulted in similar levels of fluorescence as the C-terminal tagged combination (Figure 3).

We believe that testing the role of W49 in AtEH1/Pan1 for SCAMP5 trafficking by assessing the ability of W49A mutated AtEH1/Pan1 to rescue the *ateh/pan1* male sterile mutant phenotype is not feasible. First of all, we would have to tackle AtEH1/Pan1 and AtEH2/Pan1 sequentially. Secondly, SCAMP5 mutants are viable and macroscopically similar to wild type (our unpublished data), likely due to redundancy with other SCAMP family members, some of which lack the double NPF motif. The specific role of SCAMP5 during pollen and plant development is currently unknown. Alternative pathways might step in during the absence of the interaction between

SCAMP5 and AtEH1/Pan1. If the W49A mutant of AtEH/Pan1 can rescue the pollen lethal phenotype of the *ateh/pan1* mutant, it will, in our opinion, merely address that the interaction between AtEH/Pan1 and SCAMP5 is not essential for pollen development.

Minor points: - Is the NPF motif conserved in other SCAMP proteins? is it found in other proteins?

The double NPF motif is, based on a search using the KEGG motif finder (<https://www.genome.jp/tools/motif/MOTIF2.html>), only present in a very limited number of proteins. The double NPF motif is present in three out the five members of the Arabidopsis SCAMP family (SCAMP1, SCAMP4 and SCAMP5), although all five have a single NPF motif and negatively charged residues in its vicinity (see alignment in Figure A below). Next to this family, a double NPF motif is also present in the Nucleoporin autopeptidase protein NUP98A (AT1G10390). When searched for a NPF motif preceding or followed by a negatively charged stretch, using the KEGG MOTIF finder, two t-SNARE proteins were identified, SNAP25 and SNAP33, along with two uncharacterized plasma membrane proteins. Although speculative at this stage, we believe that the nuclear envelope localization that we specifically observe in *N. benthamiana* with the 3xEH1.1-GFP construct might actually represent binding of the EH-domain to the double NPF motif of NUP98A.

Figure A: alignment of the N-terminal region of the five Arabidopsis SCAMP proteins.

Reviewer #2 (Remarks to the Author):

In this work, Yperman, et al. describe structural and biochemical analysis of two EH domains within the AtEH1/Pan1 subunit of the plant-specific TPLATE complex (TPC). The authors of the manuscript are the foremost experts in the TPLATE complex, having isolated the complex and characterized its function in several previous publications. In this work, they investigate the role of two EH domains which seem to have diverged in their specific function without losing their EH structural fold, an interesting insight into the evolutionary diversification of ancestral protein motifs. While TPC has two highly homologous AtEH subunits (AtEH1 and AtEH2), the authors focus on the two EH domains in AtEH1, which they term EH1.1 and EH1.2. Previous NMR structures of EH1.1 and EH1.2 exist and the authors build on this work by determining a high-resolution crystal structure of EH1.1 and using integrative modeling of EH1.2. They show that EH1.1 chelates a sodium ion in addition to the canonical calcium ion, whereas the EH1.2 chelates two calcium ions, a novel finding as all known EH domains chelate a single metal ion. The rest of the work focuses on understanding the diversification of function between the two domains. Using lipid binding assays, the authors propose that EH1.2 has a preference for binding PI(4,5)P₂, a lipid enriched at the plasma membrane, consistent with the role of TPC during endocytosis. Using mass spec, binding assays, and fluorescence microscopy, they show that EH1.1 interacts with the SCAMP5 protein and demonstrate that perturbing this interaction shows disruptions in localization and trafficking consistent with the role of TPC in recycling SCAMP5 from the plasma membrane. I think the model is quite intriguing that a single protein fold has been diversified in TPC to lend localization/lipid binding specificity to one domain and cargo recruitment to another.

Overall, the manuscript is well written and the figures and data presentation are superb. Much of the work is excellent and very neatly supports their proposed models. They present many redundant forms of evidence for the metal binding capacity of the two domains and several experiments investigating the SCAMP5 interaction. This work is all technically sound and I have very few comments below in the “minor criticisms” sections. My main criticism is in regard to the lipid binding profile of the two domains. This is a central finding of the work and needs to be addressed before publication.

Major criticisms that need addressing:

The major potential issue in this work is that one of the primary results, namely that EH1.2 has a preference for PI(4,5)P₂ binding, is not well supported by their data and in my opinion the two assays they used to come to this finding show contradictory results. The authors claim that EH1.2 and not EH1.1 has a preference for PI(4,5)P₂ and that this is perturbed by calcium concentration, suggesting a charged ionic interaction. Consistent with this, EH1.2 has a “basic patch” that is lacking in EH1.1.

In Supplemental Figure 4b, the authors show binding assays using purified EH domains and PolyPiPosomes containing various lipid species (PC, PIP₃, PIP₄, PI_{4,5}P₂) that were purchased from a company. By eye, the binding preference for both EH domains seems very weak, if non-existent. Both domains appear to bind PC and all PIPs equally well, suggesting there is no discrimination by a single EH domain for a preferred lipid, or a difference in binding profile between the two EH domains. However, simply looking at the gels is not sufficient. The authors performed this experiment in triplicate and quantified the results, normalizing to the PC binding, shown in Supp Figure 4C. No statistical analysis seems to have been performed to show a significant difference between any of the various interactions, and looking at the raw values they have plotted I am not confident that any would be found. With this in mind, I don't think the PolyPiPosome data can be used to make any claims about lipid preference, and maybe even supports the interpretation that there is no lipid discrimination between the EH domains or for a single EH domain for a specific lipid. One major caveat of the above

experiments was that they were performed in the presence of 100 μ M calcium, which the authors show in another set of experiments weakens the affinity of EH1.2 for charged lipids.

We agree with the reviewer that the PolyPiPosome results presented in the first submission can be perceived as not sufficiently convincing. We started out with this experiment and later performed the liposome assays and therefore moved the PolyPiPosome data to supplement. We believe that the poor binding capacity originates in the cross-linked nature of the lipids of the PolyPiPosome. This will limit the lateral movement of the lipids and if the EH domains require multiple phosphate groups for a stable binding (as suggested by our new data obtained by the extensive molecular dynamics simulations, Figure 3 and Supplemental figure 4), this will result in a reduced binding strength. To simplify our data, we removed the PolyPiPosomes from our study.

The results that the authors show in the main text in Figure 2 are much clearer by eye. Figure 2A shows that in 10 μ M calcium, EH1.2 binds to PI(4,5)P₂ and not PC, and EH1.1 does not bind either. Figure 2B shows that in the presence of 100 μ M calcium, EH1.2 only binds PI(4,5)P₂ and not PC or other charged PIPs (PIP3 and PIP4). Figure 2C shows that EH1.2 binding to PI(4,5)P₂ is dependent on calcium concentration. Taken on their own, the results in the main figures seem quite clear and support the claim that EH1.2 and not EH1.1 has a preference for the plasma-membrane enriched PI(4,5)P₂. However, these experiments do not seem to be done in triplicate or replicated in the presented work. Considering that the PolyPiPosome assays seem to have variance gel-to-gel, I don't think that a single gel can be taken to support the conclusion of the lipid binding findings.

To answer the comment of the reviewer, we now included multiple replicates of the liposome experiments backed up by statistical analysis. Our ANOVA evaluation shows significant differences in the lipid binding capacities of EH1.1 versus EH1.2, with EH1.2 binding more efficiently to PI(4,5)P₂ than EH1.1 (Figure 2 panel a and b).

We also added independent replicas of the lipid assay showing EH1.2 interaction with other phosphoinositides (Supplementary figure 3a and b) together with independent replicas showing an effect of calcium on the interaction between EH1.2 and PI(4,5)P₂ (Supplementary figure 3c). In addition, we included new data and the quantification on the interaction of both EH domains with phosphatidic acid (PA), together with new data obtained with the EH1.2 domain mutated in its predicted lipid binding site (Figure 2 and Supplementary figure 3). We believe that the new experiments, repetitions and statistical analysis convincingly support our claims.

As the authors note, perturbing PI(4,5)P₂ levels in planta is not feasible. They use multiple methods to show that PIP4 does not drive AtEH1 localization (Fig 2F-H), but I believe this data has to stand on its own and can't be used to justify any claims about PI(4,5)P₂ binding.

We agree with the reviewer that the PAO data (i.e. affecting PI4P levels *in planta*) does not allow us to make any claims on AtEH1/Pan1 binding to PI(4,5)P₂. Therefore, we decided to remove the data from the current version of the manuscript. To address membrane recruitment *in planta*, we expressed EH1.1 and EH1.2 as a triple repeat in tandem orientation fused C-terminally with GFP in *N. benthamiana* leaves. Co-expression of the three-tandem domains together with cyclin D, which allows to monitor cell plates in this system, resulted in a clear localization to the membrane of the cell plate in case of EH1.2 and a more persistent labeling of the cell plate as well as a labeling of the nuclear envelope in case of EH1.1. We hypothesize the latter to be caused by a coincidence detection of lipids and protein interactions (Figure 2). We have discussed this also in the current version of the manuscript.

On page 4, the authors claim "PolyPiPosome assays confirmed our findings (Supplementary Figure 4)...". I do not agree with this assertion and, on the contrary, I think that data contradicts the findings from the main figures.

As stated above, we replaced the PolyPiPosome assays in our current version by additional replicas and statistical analysis of our liposome data.

Minor criticisms:

- In Supplementary figure 1, panel e, there is a region of positive difference density that is close to the bound calcium ion. I could not find a reference to this in the text. Is this an unmodeled water? A bound molecule from the mother liquor?

The positive difference density is located between the Ca^{2+} ion and the aspartic acid (see Figure B below) and cannot readily be interpreted as an additional coordinating water because this would cause major clashes. The observed difference density is most likely due to a suboptimal description of the atomic displacement parameters for the Ca^{2+} ion. At 1.55 Å resolution the data to parameter ratio is not favorable enough to allow for a robust refinement of individual anisotropic B-factors. Instead, the anisotropic thermal motion of the non-solvent atoms was described by 9 TLS groups. The Ca^{2+} atom was accommodated in the TLS group encompassing most of the residues coordinating to the calcium atom (residues 21 through 29). This approach led to a better fit of our model to the data.

Figure B: different orientation of supplemental figure 1e showing that the density is located between the calcium and the aspartic acid.

- In Figure 3f the authors describe the colocalization of their proteins of interest during different phases of the cell cytokinesis.

“Both proteins also prominently co-localize during various stages of cell plate formation where SCAMP5 clearly precedes the arrival of AtEH1/Pan1. However, the presence of AtEH1/Pan1 at the newly formed cross wall exceeds SCAMP5 following completion of cytokinesis (Figure 3, panel f)”
As non-experts in plant biology, it took us a good deal of concentration to understand the figure. I think labels for the cells as “early” “mid” or “late/post-cytokinesis” would be helpful. At first glance it looks like multiple examples of the same phenomenon but from the legend it seems that the panels left to right are meant to imply a temporal sequence of cell plate growth throughout cytokinesis.

Indeed, there is a temporal sequence in cell plate formation, secretion initiates cell plate formation and endocytosis only kicks in at a certain point. We wanted to show that SCAMP5 arrives during the early phases, before the endocytic machinery is recruited. We thank the reviewer for the suggestion. As a solution, we added labels to indicate early, mid and post cytokinesis to Figure 3 and we made this clearer in the figure legend.

Reviewer #3 (Remarks to the Author):

The focus of this manuscript is the characterization of two EH domains that belong to the AtEH/Pan1 protein from Arabidopsis. AtEH/Pan1 is a subunit of the TPLATE complex that is involved in clathrin-mediated endocytosis in Arabidopsis. The authors' major conclusion [and statement of significance/novelty] is that the two EH domains, EH1.1 and EH1.2 have different functional properties: one is a sensor of the anionic phospholipid PI(4,5)P₂, and the other interacts with the NPF motif of the novel TPC interacting partner, SCAMP5. In this reviewer's opinion, there are three major problems with this manuscript:

1. Conclusions are not supported by the data. 2. The figures do not support the statements made in the text. 3. The novelty of the work is unclear, as the findings are not at all put in the context of what is known about the EH domains from other organisms (Mouse EHD2 (2QPT) was used for molecular replacement). This pertains to both structural and functional work.

Issues 1 and 2 are expanded upon below:

A. Results and Discussion section: "The second EH domain coordinates two calcium ions" The authors imply that Ca²⁺ is needed for folding. NMR spectra that they show in Fig 1 (i-j) suggest exactly the opposite – addition of 10 mM EDTA to sequester Ca²⁺ has very little effect on the protein conformation. It is therefore unclear how NMR data are used to support the folding argument.

We agree that the overall size of the panels and the colors used to overlay the spectra hinder the visualization of the drastic effect of EDTA on the structural integrity of both EH domains. We have split the main panel, now showing the spectra before and after addition of EDTA to each domain to make the effect more clear (Figure 1j). In addition, we also include the corresponding 1D spectra here (Figure C). Please notice that all spectra have been acquired with exactly the same parameters allowing for a direct comparison and conclusions to be drawn. Quantification of peak intensities for 4 methyl signals around 0 p.p.m. (a clear signature of any folded domain) shows clearly that for EH1.1 less than 50% of the protein remains in solution after 10mM EDTA addition. The rest of the protein has been incorporated in larger particles that do not yield any NMR signal due to slow tumbling in solution. For EH1.2, EDTA effects are pronounced. The protein precipitates heavily, no signals of a folded domain are present (methyl signals not present around 0 p.p.m.) only signals of flexible residues remain that correspond to large soluble aggregates.

Figure C: 1D spectra of EH1.1 and EH1.2 before (black) and after (red) the addition of 10mM EDTA.

We agree with the reviewer that our NMR data does not allow us to differentiate between unfolding and aggregation and in order to avoid data misinterpretation, we removed Supplementary figure 3 from our manuscript and adapted the text.

Neither MD simulations nor sensitivity of EH1.2 to precipitation upon sequestration of Ca^{2+} by EDTA can be used to validate the Ca^{2+} binding stoichiometry arguments. The stoichiometry argument (EH1.2 binding two Ca^{2+} ions, not one) – especially if the authors present this as a novel finding – must be validated using direct methods, such as, e.g., inductively coupled plasma measurements.

To confirm our total reflection x-ray fluorescence (TXRF) data, as suggested by the reviewer, we have now also performed ICP-MS, which showed the presence of two bound calcium ions in the case of EH1.2 and we have added these results to Figure 1 (Figure 1i).

The NMR structure of EH1.2 appears to be calculated with just one Ca^{2+} ion, and then the second Ca^{2+} ion is added based on the MD results; this structure is presented in Fig 1e. This reviewer questions the validity of this procedure and representation of the structural data, especially given that the Ca^{2+} stoichiometry must be established experimentally using direct methods (see above).

Indeed, the NMR structure of EH1.2 was determined with just one calcium atom. As stated in Material and Methods, we included calcium in a pentagonal bipyramidal coordination for the canonical EF-hand motifs, one in EH1.1 and one in the EH1.2 domain. Both NMR structure ensembles are shown in Supplemental figure 1f-g. In the figure legend we have added the following sentence “One calcium atom was included in each structure based on the coordination observed in canonical EF-hand motifs” to make clear how we determined the NMR structures. From the TXRF and ICP-MS data we however know that EH1.2 accommodates a second calcium atom but in the second EF-hand motif the Glu residue is replaced by Gln. Searching the PDB database we could not find a crystal structure where the conserved Glu residue of the EF-hand motif is replaced by Gln to get insight on the alternative calcium coordination. Therefore, we turned to all-atom MD simulations seeking a plausible coordination scheme for the non-canonical motif.

In the Material and Methods, where we describe the MD simulations, we now state that we started from the NMR structure that contained one calcium atom for the canonical motif and added a second calcium atom asking whether it will be stably incorporated in the non-canonical motif and if so, what is then the stable coordination scheme. The resulting structure is displayed in figure 1e where we state that it is an “NMR/all-atom molecular dynamics structure” to reflect the procedure we followed. In the original version of our manuscript, we also phrased very carefully the wording in the main text to make clear that the non-canonical coordination displayed in Figure 1g is a prediction and not based on experimental evidence.

In addition, we performed a post-analysis comparing the starting NMR structure with five MD structures derived from the last 200 ns of the all-atom MD simulation (Supplementary figure 2c). Two important observations became apparent. First, there is no need for large structural perturbations to incorporate the second calcium atom, overall RMSD is 2Å and the key residues in both motifs, canonical and non-canonical, are perturbed to the same extent. Second, the MD simulations converge to a stable coordination that resembles the starting conformation of the residues involved (Supplementary figure 2d). In principle, we think that our approach is a valid procedure that provides a model for the coordination of a second calcium atom that was experimentally determined by TXRF and ICP-MS.

Finally the authors make an argument that Ca^{2+} is required for folding and issue the following statement: “The ability to unfold and refold, relating to a non-functional versus a functional state, in a calcium-

dependent manner, hints at a modulatory role for calcium to control the function of this domain.” The precipitation that they observe upon removing Ca^{2+} can be a consequence of protein aggregation without loss of fold. The gel-filtration data authors present can also indicate aggregation without unfolding. If the authors wish to make an argument about the loss of fold, they should provide direct evidence that the protein loses the tertiary and secondary structure upon Ca^{2+} removal.

As we already wrote above, we agree with the reviewer that we cannot differentiate between unfolding and aggregation. In order to avoid data misinterpretation, we removed the corresponding text and Supplementary figure 3 from our current manuscript.

B. Results and Discussion section

"The second EH domain interacts with charged lipids"

All lanes in liposome-binding experiments appear essentially identical. This applies to three sets of data shown in Fig. 2a,b, and e. This reviewer cannot find any support for the statements issued by the authors:

“Only EH1.2 bound to $\text{PI}(4,5)\text{P}_2$ enriched liposomes (Figure 2, panel a).”

“We observed a very weak interaction with PI3P and PI4P compared to $\text{PI}(4,5)\text{P}_2$ liposomes (Figure 2, panel b).”

“We observed increased binding of EH1.2 at lower calcium concentrations (Figure 2, panel e).”

We now included multiple replicates of the liposome experiments to show the significant difference in the lipid binding, backed up by statistical analysis (Figure 2 and Supplementary figure 3). We also included new data on the interaction of both EH domains with phosphatidic acid (PA) together with new data obtained with the EH1.2 domain mutated in its predicted lipid binding site (Figure 2 and Supplementary figure 4).

C. Results and Discussion section "The first EH domain recognizes a novel retrograde transport motif"

The authors state “This peptide also showed the highest fold change compared to EH1.2 and GFP (Figure 3, panel a)”. Figure 3a shows a cartoon representation of SCAMP5 but no data.

We thank the reviewer for indicating this error in the text. This data refers to the mass spectrometry list, which is added as a supplement (Supplemental data file 3). We made this clear in the text.

The authors demonstrate that both domains EH1.1 and EH1.2 (Figure S6) bind the NPF peptides with about 6-fold difference in affinities. Yet they somehow conclude that only EH1.1 recognizes the retrograde transport motif. This conclusion is not supported by the data.

Due to the vast difference between the binding affinities based on NMR (30 vs 190 μM), the fact that we could identify the peptide in our proteomics approach only with EH1.1 as well as the inability to detect binding between the peptide and EH1.2 via BLI we stated that EH1.1 is the NPF-binding partner. We further clarified in the text that both EH domains showed the ability to bind double NPF peptides, but that the binding capacity of EH1.1 exceeded that of EH1.2 and

that we, therefore, find it most likely that this domain will preferentially bind the double NPF motif of SCAMP5.

We believe that the novelty of our paper lies in the fact that this type of analysis, where we directly compare both EH domains, *in vitro* as well as *in planta*, of a single protein has not been performed yet. Moreover, although EH domains have been investigated structurally and functionally in other kingdoms, this is not the case in plants. To our knowledge, there are also no reports of EH domains binding two calcium ions.

As suggested by the reviewer, we have edited our manuscript and included a part where we compare our findings better in the light of what is known about EH domains in other organisms.

Reviewer #4 (Remarks to the Author):

Comments to the Author

This manuscript reported the structural and functional roles of the evolutionary conserved N-terminal Eps15 homology (EH) domains of the TPC subunit AtEH1/Pan1 in Arabidopsis, which is involved in Clathrin-mediated endocytosis (CME) process. The authors integrated various experimental techniques including X-ray crystallography and NMR spectroscopy with all-atom molecular dynamics simulations, to discover that the first EH domain binds SCAMP5 (a novel TPC interactor) by interacting with the N-terminal double NPF motif of the latter; the second EH domain interacts with negatively charged phospholipids PI(4,5)P₂. The complementary roles of the EH domains of AtEH/Pan1 in plant addressed in this manuscript would provide insights into the understanding of CME process in plant. I would like to recommend this manuscript to be published in Nature Communications after a revision.

Specific comments and questions:

1. The Authors claimed that “The all-atom molecular dynamics model, within its time and force field limitations, suggests that the first aspartate and the presence of an extra water molecule compensate for the incomplete calcium-binding motif in the first loop of EH1.2 and functionally mimics the role of the canonical glutamate in the calcium-binding motif” , but they did not provide any evidence, such as structure snapshot in the simulated trajectories or other statistical information from the trajectories for EH1.2 WT to support it. Meanwhile, it seems the simulations were performed in explicit solvent, then, what the water model used in the simulations should be addressed.

As the reviewer rightfully points out, the simulations were performed in explicit solvent. The water model used in the simulations was TIP3P. We made this clear in the Material and Methods and provided an additional analysis showing presence of two water molecules in loop 1 of EH1.2 over the last 500 ns of the all-atom MD simulation (Supplementary figure 2e and f). To further clarify the coordination of water in both EH1.2 loops, we adapted the cartoon representations in the main figure to include the water molecules as obtained by the all-atom MD simulation (Figure 1f and g).

2. Supporting Figure 5 was not mentioned in the main text. Further, the specific panel(s) of Supporting Figure should be clarified to make the manuscript more readable.

We thank the reviewer for the comment. We made the required changes to make the references to the figures more clear.

3. There are some errors or conflicts in the manuscript: (1) In the first paragraph of page 3, should “With respect to EH1.2, the NMR and X-ray structures agree very well with an RMSD of 1Å” be “With respect to EH1.1,” ?

(2) In the first paragraph of page 4, should “The lipid interacting residues in the EH domains of EHD1 and Eps15 are structurally conserved in EH1.2 (K391 and K398), we hypothesize a third residue, K384,” be “EH1.2 (R391 and K398)R384”? Or else, it is not consistent with Figure 2c and 2d. Meanwhile, in Figure 2d, for EH1.2, the label “R398” should be “K398”.

(3) The color scale in Figure 3g is not consisted with that in Supporting Figure 7a.

We thank the reviewer for pointing out these mistakes. We made the required changes.

Reviewers' comments:

Reviewer #1 (Remarks to the Author):

I thank the authors for extensively addressing my comments through new experiments/discussions. I really like the new data on PA binding and the idea to use cyclinD to observe cell plate localization in transiently transfected *N. benthamiana* leaf. The rBIFC data, which include the W>A non-binding mutant, are also convincing.

I still think it would be interesting to test the localization of an AtEH1/pan1 mutated on Lysines 384, 391 and 398 to address the importance of PIP2 binding on the localization of this subunit. Perhaps it is compensated by other lipid-interacting subunits or by residual PA-binding in AtEH1/pan1, but maybe not, only such experiments can tell. Ultimately, the functional relevance of PIP2 binding by AtEH1/pan1 remains to be determined in my opinion. In any case, I understand that this could be a time-consuming experiment and I acknowledge that the manuscript stands on its own as is, with a clear structural rather than functional focus. I thus support its publication.

Reviewer #2 (Remarks to the Author):

In my original review, my only major concern was the data regarding the lipid binding profile of the two EH domains. The authors have addressed this and have also included new data about the lipid phosphatidic acid. They have also made other changes to address the concerns of the other reviewers. I have no new comments for this manuscript and recommend acceptance/ publication.

Reviewer #4 (Remarks to the Author):

Comments to the Author

In the revised manuscript (NCOMMS-20-20224A-Z), the authors added more experimental and simulation data as well as interpretations, meanwhile removed some contents to support their conclusion. However, some of the complementary simulation data were not clearly addressed and some seem not supportive.

(1) The RMSD of individual residues in Supplementary Figure 2 (c) could not be used to assess conformation similarity with that from NMR experiment. I suggest RMSD calculation of the loop which is coordinated with Ca²⁺ might be better.

(2) In Supplementary Figure 2 (e) and (f), it would be more clear and direct to record the number of oxygen atom from water molecule which is coordinated with Ca²⁺ versus simulation time.

(3) Supplementary Figure 4 (c) (d) could not provide any meaningful information due to so many replicas with different color massed together, and I don't understand what the authors want to address by presenting "the minimal distance between the center of mass of EH domain and the center of mass of the lipid bilayer" because it does not correspond to the closest distance of the two parts (the EH domain might not bind the center position of the lipid bilayer), meanwhile, what does the average distance from those independent replicas (vs simulation time) mean?

(4) For Fig. 2(f) and (g), the author did not address clearly how the contact was calculated, through all trajectories or through the time regions that binding is stabilized? Because from Supplementary Figure 4 (e)-(f), it is obvious that the EH domain might collide lipid layer many times before it binds the latter stably, which means some contacts in the whole trajectory might not contribute to the binding process.

(5) The conclusion in main-text "Our data also suggest that once EH1.2 establishes long-lived interactions with PA, it remains stably bound to the PA-containing membrane. This was not the case for EH1.1." are not supported by Supplementary Figure 4 (e)-(h) solidly due to the statistical analysis is too roughly.

(6) Some figures and legends were not checked carefully. For example, in Supplementary Figure 2 (b), there are no purple and orange lines; the residue numbers presented in (b) and (c) are different; etc.

REVIEWER COMMENTS

Reviewer #1 (Remarks to the Author):

I thank the authors for extensively addressing my comments through new experiments/discussions. I really like the new data on PA binding and the idea to use cyclinD to observe cell plate localization in transiently transfected *N. benthamiana* leaf. The rBIFC data, which include the W>A non-binding mutant, are also convincing.

I still think it would be interesting to test the localization of an AtEH1/pan1 mutated on Lysines 384, 391 and 398 to address the importance of PIP2 binding on the localization of this subunit. Perhaps it is compensated by other lipid-interacting subunits or by residual PA-binding in AtEH1/pan1, but maybe not, only such experiments can tell. Ultimately, the functional relevance of PIP2 binding by AtEH1/pan1 remains to be determined in my opinion. In any case, I understand that this could be a time-consuming experiment and I acknowledge that the manuscript stands on its own as is, with a clear structural rather than functional focus. I thus support its publication.

We thank the reviewer for supporting our work.

Reviewer #2 (Remarks to the Author):

In my original review, my only major concern was the data regarding the lipid binding profile of the two EH domains. The authors have addressed this and have also included new data about the lipid phosphatidic acid. They have also made other changes to address the concerns of the other reviewers. I have no new comments for this manuscript and recommend acceptance/publication.

We thank the reviewer for supporting our work.

Reviewer #4 (Remarks to the Author):

Comments to the Author

In the revised manuscript (NCOMMS-20-20224A-Z), the authors added more experimental and simulation data as well as interpretations, meanwhile removed some

contents to support their conclusion. However, some of the complementary simulation data were not clearly addressed and some seem not supportive.

We thank the reviewer for pointing out some remaining items in our MD simulations to further improve the quality of our work. We have now addressed them and hope that our work now meets the reviewers' criteria to be accepted for publication.

(1) The RMSD of individual residues in Supplementary Figure 2 (c) could not be used to assess conformation similarity with that from NMR experiment. I suggest RMSD calculation of the loop which is coordinated with Ca²⁺ might be better.

In order to address the suggested change, we calculated the respective RMSD values between the NMR structure and five timepoints within the last 200 ns of the MD simulation for each EF-hand loop. We added these values to Supplementary figure 2, panel c. We also marked both loops in that figure and marked the amino acid residues of both loops to allow for better assessment of the conformation similarity between the NMR and the MD simulation. Our calculated RMSD values for both loops are around 2 Å corresponding to the overall RMSD. This result indicates that both loops are perturbed to the same extent and there is no need for large structural perturbations of loop 1 to incorporate the calcium atom.

(2) In Supplementary Figure 2 (e) and (f), it would be more clear and direct to record the number of oxygen atom from water molecule which is coordinated with Ca²⁺ versus simulation time.

We thank the reviewer for the suggestion. We now included two additional figures (Supplemental figure 2, panel e and f) where we show the number of oxygen atoms from water that coordinate calcium in each EF-hand loop of EH1.2 over the complete duration of the all-atom MD simulation. From the new plots in panels e and f, it is now clear that on average over the length of the simulation time, loop 1 uses an extra oxygen from water to coordinate the Ca²⁺ compared to loop 2.

(3) Supplementary Figure 4 (c) (d) could not provide any meaningful information due to so many replicas with different color massed together, and I don't understand what the authors want to address by presenting "the minimal distance between the center

of mass of EH domain and the center of mass of the lipid bilayer” because it does not correspond to the closest distance of the two parts (the EH domain might not bind the center position of the lipid bilayer), meanwhile, what does the average distance from those independent replicas (vs simulation time) mean?

We agree with the reviewer that the individual replicas in the original Supplemental figure 4 panel c and d are difficult to discern.

We therefore decided to remove these two panels as the information is also displayed in the original panels e and f. In this way, we also solve the issue the reviewer has with the average distance of the independent replicas, which was merely a way to present the general trend in the simulation.

The previous panels e and f are now panels c and d.

The minimal distance between the two centers of mass that the reviewer has raised is a textual mistake. We apologize for this and we would like to thank the reviewer for noticing it. We have corrected the text in the manuscript, accordingly (e.g. the figure legend of Supplementary figure 4 now states that the progress of the simulations is shown as the minimal distance between the protein and the lipid bilayer).

The distance depicted in the various panels in the current Supplementary figure 4 always represents the minimum distance between any pair of atoms from the lipid bilayer and the protein molecule. We now clearly state this in the legend.

(4) For Fig. 2(f) and (g), the author did not address clearly how the contact was calculated, through all trajectories or through the time regions that binding is stabilized? Because from Supplementary Figure 4 (e)-(f), it is obvious that the EH domain might collide lipid layer many times before it binds the latter stably, which means some contacts in the whole trajectory might not contribute to the binding process.

The contacts were calculated through all trajectories. We believe that this is a reasonable approximation as we wanted to compare those residues with the highest number of contacts that contribute to the binding process for each domain. We believe that even if some contacts result from random collisions of the protein with the membrane, they would not significantly contribute to the number of contacts observed. We clarified now in the text that we calculated the contacts through all trajectories (CG-MD replicas).

(5) The conclusion in main-text “Our data also suggest that once EH1.2 establishes long-lived interactions with PA, it remains stably bound to the PA-containing membrane. This was not the case for EH1.1.” are not supported by Supplementary Figure 4 (e)-(h) solidly due to the statistical analysis is too roughly.

To meet the concern of the reviewer and to support our conclusions better, we rephrased the text and performed an additional analysis of our CG-MD simulations. We calculated the number of PA molecules in proximity of each EH domain over the two μ s-long MD simulations. The analysis showed that EH1.2, on average, coordinates more PA molecules compared to EH1.1. This supports our biochemical data that EH1.2 binds PA more efficiently than EH1.1. We included this analysis as a part of Supplementary figure 4 (panels g and h) and adapted the text accordingly.

(6) Some figures and legends were not checked carefully. For example, in Supplementary Figure 2 (b), there are no purple and orange lines; the residue numbers presented in (b) and (c) are different; etc.

We thank the reviewer for noticing these mistakes. We adapted the text and figures accordingly.

Reviewers' comments:

Reviewer #4 (Remarks to the Author):

The authors have addressed all my concerns, and I have no new comments for this manuscript and support its publication.